# DATA VALUATION BY LEVERAGING GLOBAL AND LOCAL STATISTICAL INFORMATION

## ABSTRACT

Data valuation has garnered increasing attention in recent years, given the critical role of high-quality data in various applications. Among diverse data valuation approaches, Shapley value-based methods are predominant due to their strong theoretical grounding. However, the exact computation of Shapley values is often computationally prohibitive, prompting the development of numerous approximation techniques. Despite notable advancements, existing methods generally neglect the incorporation of value distribution information and fail to account for dynamic data conditions, thereby compromising their performance and application potential. In this paper, we highlight the crucial role of both global and local statistical properties of value distributions in the context of data valuation for machine learning. First, we conduct a comprehensive analysis of these distributions across various simulated and real-world datasets, uncovering valuable insights and key patterns. Second, we propose an enhanced data valuation method that integrates the explored distribution characteristics into two regularization terms to refine Shapley value estimation. The proposed regularizers can be seamlessly incorporated into various existing data valuation methods. Third, we introduce a novel approach for dynamic data valuation that infers updated data values without recomputing Shapley values, thereby significantly improving computational efficiency. Extensive experiments have been conducted across a range of tasks, including Shapley value estimation, value-based data addition and removal, mislabeled data detection, and dynamic data valuation. The results showcase the consistent effectiveness and efficiency of our proposed methodologies, affirming the significant potential of global and local value distributions in data valuation.

## 1 INTRODUCTION

Data valuation aims to quantify the value of a datum in a dataset for various applications, including business decision-making, scientific discovery, and model training in machine learning (Ai et al., 2025; Garrido Lucero et al., 2025). It is a rapidly evolving and high-impact research topic in data-centric research communities and industrial areas, as a dataset with a large proportion of highly valuable data greatly benefits real applications (Guo et al., 2025; Wen et al., 2025; Pei, 2022). Existing data valuation methods can be broadly categorized into four groups (Jiang et al., 2023): marginal contribution-based (Kwon & Zou, 2022; Luo et al., 2024; Garrido Lucero et al., 2025), gradient-based (Koh & Liang, 2017; Just et al., 2023), importance weight-based (Yoon et al., 2020), and out-of-bag estimation-based (Kwon & Zou, 2023) methods. Among these, the marginal contribution-based approach has emerged as the most popular and delivers strong performance. This method quantifies a datum's value by assessing the average change in utility when the datum is removed from a set of fixed cardinality.

An important index, namely, the Shapley value, which is a key concept in cooperative game (Winter, 2002; Li et al., 2024), is usually utilized to calculate the marginal contribution for data valuation. Due to its solid theoretical basis, Shapley value is among the primary choices in data valuation (Stoian, 2023; Luo et al., 2024; Garrido Lucero et al., 2025). However, the accurate calculation of the Shapley value for a given data corpus is nearly intractable as the computational complexity is about $O(2^N)$ for $N$ samples. Therefore, researchers have made efforts toward an approximate yet efficient valuation methodology. For example, Jia et al. (2019) investigated the scenario when data are employed for training a KNN classifier and proposed a novel efficient method, KNN Shapley, exactly

in $O(N \log N)$ time. Moreover, a recent study introduces a sparsity assumption on data values to alleviate the computational burden associated with an approximate method, specifically the Average Marginal Estimation (AME) approach (Lin et al., 2022).

Although promising results are obtained, we argue that ***the potential of value distribution in data valuation has been largely neglected in nearly all previous studies***. The sparse assumption utilized in AME actually presumes that the data values in a dataset conform to the Laplace distribution (detailed in Section 2). However, our findings indicate that this assumption may not always be justified. The value distribution in this study consists of two parts: local distribution, which captures the relationship between a datum and its neighborhood, and global value distribution for all involved data. Through our empirical analysis, we have observed that the distribution of data values in a dataset more closely follows a Gaussian distribution rather than a Laplace distribution. Furthermore, our findings indicate a strong correlation among the values of nearby samples (i.e., samples within the same neighborhood). Specifically, the similarity in values between neighboring instances within the same category is pronounced, whereas the similarity between neighboring samples from different categories is minimal.

Another key motivation is dynamic data valuation, which involves quantifying data values in scenarios where new data is introduced or existing data is removed. To the best of our knowledge, only one existing study tries to address dynamic data valuation (Zhang et al., 2023). This pioneering work adapts the traditional Shapley value calculation into an incremental paradigm, achieving a significant reduction in computational cost—up to half—when adding or removing a datum. Building on our earlier observations where the value of an individual datum can be inferred from its surrounding neighborhood, we are inspired to explore an alternative approach to dynamic data valuation.

This study investigates both the global and local distribution characteristics of data values and explores how these characteristics can be applied to both conventional and dynamic data valuation methods. First, various synthetic and real datasets are leveraged to make statistical analyses of the characteristics of global and local value distributions. Useful observations and clues are obtained on the basis of the statistical results and the discussion of previous methods. Second, two new methods for data valuation are proposed. Specifically, the first method applies the distribution characteristics to one classical Shapley value-based data valuation method, namely, AME (Lin et al., 2022). Many existing methods can replace AME in our approach. The second method introduces a novel optimization problem that integrates distributional characteristics for dynamic data valuation, eliminating the need to re-estimate the Shapley values of the data, thus significantly improving efficiency. Third, comprehensive experiments are conducted on various benchmark datasets to evaluate the effectiveness of our methodologies in data valuation across a range of tasks.

The experimental results on Shapley value estimation indicate that, compared to the AME approach, our method provides a more accurate approximation of the true Shapley values. Moreover, experiments on value-based point addition and removal tasks demonstrate the effectiveness of our approach in identifying both influential and poisoned samples. Furthermore, our method outperforms other data valuation techniques in mislabeled data detection tasks. Additionally, the proposed dynamic data valuation approaches consistently achieve state-of-the-art performance while significantly enhancing computational efficiency.

## 2 RELATED WORK

**Data Valuation.** High-quality data play a crucial role in numerous real-world applications (Fleckenstein et al., 2023; Sim et al., 2022; Guo et al., 2025). However, real-world datasets often exhibit heterogeneity and noise (Liang & Zou, 2022; Sun et al., 2024). Therefore, accurately quantifying the value of each datum within a dataset is essential for various applications and data transactions in the data market. As discussed in Section 1, existing data valuation methods can be broadly categorized into four main types:

- **Marginal contribution-based methods**: This kind of method calculates the differences of the utility with or without the datum to be quantified. The larger the utility difference is, the more valuable the datum is. Representative methods include leave-one-out (LOO) (Jiang et al., 2023), Data Banzhaf (Wang & Jia, 2023), and a series of Shapley value-based methods such as Du-shapley (Garrido Lucero et al., 2025), Beta Shapley (Kwon & Zou, 2022), and AME (Lin et al., 2022).

- **Gradient-based methods**: This kind of method evaluates the change in utility when the weight of the datum under assessment is increased. Two representative methods are Influence Function (Feldman & Zhang, 2020) and LAVA (Just et al., 2023).

- **Importance weight-based methods**: This kind of method learns an important weight for a datum to be quantified during training and takes the weight as the value (Fleckenstein et al., 2023). Naturally, importance weight-based methods are particularly proposed for machine learning applications. One representative method is DVRL (Yoon et al., 2020), which utilizes the reinforcement learning technique to learn sample weights.

- **Out-of-bag estimation-based methods**: This kind of method is also designed particularly for machine learning tasks (Sim et al., 2022). The representative method, Data-OOB (Kwon & Zou, 2023), calculates the contribution of each data point using out-of-bag accuracy when a bagging model (e.g., random forest) is employed.

Additionally, Jiang et al. (2023) developed a standardized benchmarking system for data valuation. They summarized four downstream machine learning tasks for evaluating the values estimated by different data valuation methods. Their results suggest that no single algorithm performs uniformly best across all tasks. Moreover, Zhang et al. (2023) proposed an efficient updating method for dynamically adding or deleting data points. In their study, three specific algorithms are introduced, which reduce the overall computational cost compared to previous Shapley value-based methods. However, existing algorithms largely ignore the distributional information of data values, resulting in suboptimal performance and efficiency. Incorporating both global and local distribution characteristics allows capturing structural relationships among samples, assessing their importance and representativeness, and guiding model training and regularization. These aspects have been applied in various machine learning tasks and are discussed in the following subsection.

**Distribution-Aided Learning.** In machine learning, both global and local distributional information have been leveraged during model training (Shah & Shukla, 2017; Zhang et al., 2018). Global distribution captures assumptions about the overall data distribution (e.g., Gaussian or Laplace), while local distribution characterizes sample relationships within neighborhoods. Two prominent methods, Lasso (Guo et al., 2023) and Ridge Hoerl (2020) regression, incorporate prior distributions of model parameters in the context of regression. Take Lasso as an example, it learns the model by solving $\min_{\boldsymbol{\omega}} \sum_{\boldsymbol{x}} ||y - \boldsymbol{\omega}^T \boldsymbol{x}||_2^2 + \lambda ||\boldsymbol{\omega}||_1$, where $\boldsymbol{\omega}$ is the model parameter, $\boldsymbol{x}$ is a sample, $y$ is the target, and $\lambda$ is a hyperparameter that controls the strength of the regularization. Lasso can be inferred from a statistical view. Assuming that the prior distribution $\boldsymbol{\omega}$ conforms to a Laplace distribution as follows:

$$\boldsymbol{\omega} \sim \frac{1}{2\sigma} \exp(-\frac{||\boldsymbol{\omega}||_1}{\sigma}), \tag{1}$$

where $\sigma$ is a parameter. When the maximum a posteriori estimation is applied, we obtain

$$\boldsymbol{\omega}^* = \arg \max_{\boldsymbol{\omega}} \ln[\prod_{\boldsymbol{x}} \frac{1}{\sqrt{2\pi}\sigma_1} \exp(\frac{-||y - \boldsymbol{\omega}^T \boldsymbol{x}||_2^2}{2\sigma_1^2}) \cdot \frac{1}{\sigma_2} \exp(-\frac{||\boldsymbol{\omega}||_1}{\sigma_2})]$$

$$\sim \arg \min_{\boldsymbol{\omega}} ||y - \boldsymbol{\omega}^T \boldsymbol{x}||_2^2 + \frac{2\sigma_1^2}{\sigma_2} ||\boldsymbol{\omega}||_1, \tag{2}$$

where the coefficient $2\sigma_1^2/\sigma_2$ can be reduced to a single hyperparameter $\lambda$. The loss in Eq. (2) is exactly the loss in Lasso. If the distribution in Eq. (1) is replaced by the Gaussian distribution, then Ridge regression can be obtained. Additionally, in multi-task learning, the distribution of the model parameters is also utilized to connect the multiple tasks. A widely used regularizer (Evgeniou & Pontil, 2004) is $\sum_t ||\boldsymbol{\omega}_t - \bar{\boldsymbol{\omega}}||_2^2$, where $\boldsymbol{\omega}_t$ is the model parameter of the $t$th task and $\bar{\boldsymbol{\omega}}$ is the mean of the model parameters. This underlying assumption for this regularizer is that $\boldsymbol{\omega}_t$ conforms to a Gaussian distribution with the mean $\bar{\boldsymbol{\omega}}$.

Local distribution is also widely utilized in various machine learning tasks. Most local distribution information refers to the high similarity between samples that are close to each other. For example, samples in the neighborhood usually share the same labels in statistics. Therefore, a well-known yet effective classifier, KNN (Peterson, 2009), is developed. Moreover, Zhu et al. (2022) designed a new linear discriminative analysis method to seek the projected directions, making sure that the within-neighborhood scatter is as small as possible and the between-neighborhood scatter is as large as possible. Furthermore, Zhong et al. (2021) revealed that a DNN trained on the supervised data generates representations where a generic query and its neighbors usually share the same label. To date, local distributional information has not been sufficiently exploited in the data valuation domain.

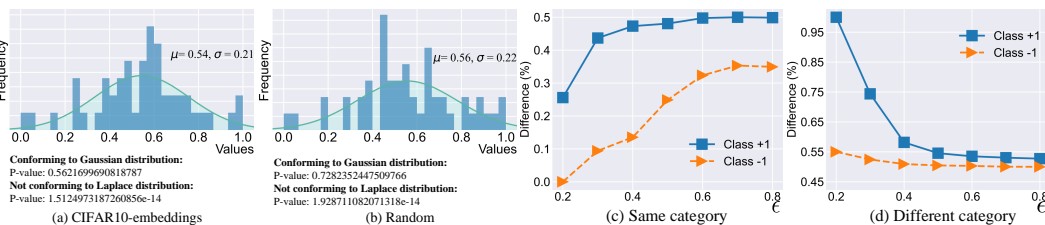

Figure 1: Distributions of data values after min-max normalization for CIFAR10-embeddings (a) and Random (b). The KStest is conducted under the null hypothesis that the data values are Gaussian distributed. The average relative difference after min-max normalization between the value of a sample and the values of its neighbors within the same (c) and different (d) categories. Random dataset defined in Eq. (12) is utilized. The blue and orange curves correspond to samples from the positive (+1) and negative (-1) classes, respectively. $\epsilon$ denotes the neighborhood range.

## 3 EMPIRICAL EXPLORATION

We conduct comprehensive analytical experiments on both simulated and real datasets to investigate the properties of global and local value distributions, as well as the changes in value when new data are added or existing data are removed. Details of the datasets used are presented in Appendix A.3.

**Analysis of Global and Local Value Distributions.** This section investigates the global and local value distributions. To estimate the Shapley values, we apply the AME-based estimator (Lin et al., 2022), setting the number of sampled subsets to the total number of samples[1]. This approach ensures that the estimated scores asymptotically converge to the true Shapley values as the number of sampled subsets is large. Two statistical analyses are performed on the estimated values. The first analysis examines the distribution of values for all samples within each dataset, while the second investigates the relative difference[2] between a sample's value and the values of its closest neighbors.

Figs.1(a) and (b) show the value distributions for two datasets: CIFAR10-embeddings and a synthetic dataset, "Random," generated using Eq. (12) in the Appendix. While these distributions resemble Gaussian or Laplace distributions, the KStest hypothesis test (Justel et al., 1997) indicates that these value distributions align more closely with a Gaussian distribution. Additional results are presented in Fig. 6 of the Appendix. These findings suggest that the value distribution is more accurately approximated by a Gaussian distribution, rather than the Laplace distribution assumed by AME.

The local characteristics of value distributions are also examined across various datasets. Specifically, we investigate the relative difference between a sample and its neighbors within the same category and across different categories. As shown in Figs. 1(c) and (d), increasing the neighborhood range, $\epsilon$, leads to an increase in the relative differences between samples within the same category and a decrease between samples from different categories. Furthermore, the relative difference within the same category is smaller compared to that between samples from different categories. These results highlight that a sample's value tends to align more closely with the values of its neighbors within the same category, with this alignment becoming stronger as the distance between samples decreases. Conversely, a sample's value shows greater divergence from the values of its neighbors from different categories, with the relative difference increasing as the distance between them decreases. We have confirmed that these observations remain consistent across various datasets.

**Analysis of Value Variations under Dynamic Data Conditions.** Two statistical analyses are conducted to examine the variation in data values when the dataset is altered. In the first analysis, 90% of the original dataset is reserved, and the AME model is applied to compute the data values for this subset. The remaining 10% of samples are then added, and new values for all data points are recalculated. The value distributions of 90% of the samples before and after adding new data are shown in Fig. 2(a). In the second analysis, 10% of the dataset is removed, and the value distributions of the remaining 90% of samples, before and after the removal, are shown in Fig. 2(b). The results indicate that while the values of the original samples exhibit some variation with the addition or removal of data, these variations are relatively small. Specifically, the changes in the mean and

---

[1]Under this setting, the sparsity assumption is not required, allowing us to use the Mean Square Error (MSE)-based estimation in AME, rather than its Lasso-based approximation.

[2]The relative difference between two values, $|a|$ and $|b|$ is calculated as $\frac{|a-b|}{\max |a|,|b|}$.

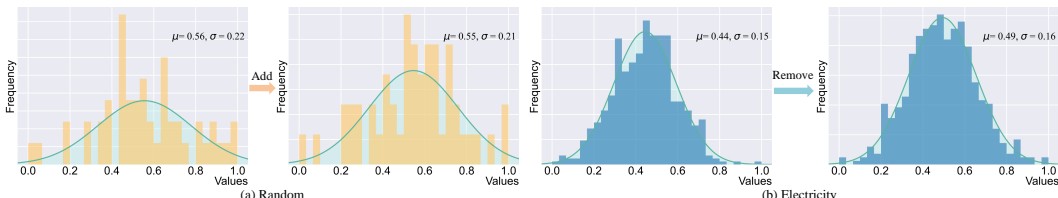

Figure 2: Variation in value distribution after adding (a) and removing partial data points (b) from the Random and Electricity datasets.

variance of the values in both datasets are less than 0.05 and 0.01, respectively. Further details on data value variation are provided in Section A.7.

**Pattern Summary.** Based on the aforementioned empirical analyses, the following observations and insights are summarized to guide the development of new methods:

- The distribution of data values across the entire dataset is found to more closely resemble a Gaussian distribution rather than a Laplace distribution. Therefore, in this study, the Gaussian distribution is adopted as the prior for data values.

- The similarity in values between adjacent samples within the same category is substantial, while the similarity between adjacent samples from different categories is minimal.

- When new data are added or existing data are removed from the original dataset, the values of samples experience changes, though these variations are relatively minor in magnitude.

In the following section, these three summarized conclusions will serve as the foundational principles for developing our new data valuation methods.

## 4 METHODOLOGY

Our approach leverages the AME method as a case study to illustrate the application of both global and local distribution information in data valuation. In Section 4.4, we further investigate how the integration of global and local value distribution information can be extended to other data valuation methods. We begin by presenting a concise overview of the AME method.

### 4.1 REVISITING THE AME APPROACH

AME is a representative data valuation approach based on marginal contributions. It begins by sampling multiple subsets from the original dataset and utilizes the performance (e.g., classification accuracy) of models trained on each subset as the utility measure. If $\mathcal{M}$ subsets are compiled, then $\mathcal{M}$ models will be trained, resulting in $\mathcal{M}$ corresponding utility scores. For each model, an $N$-dimensional feature vector is constructed to represent the composition of the training subset, where $N$ denotes the total size of the training dataset. Specifically, the $i$-th dimension of the feature vector for the $m$-th model, denoted as $\boldsymbol{X}_{m,i}$, is defined as follows: $\boldsymbol{X}_{m,i} = \frac{1}{\sqrt{vp}}$ if $\boldsymbol{x}_i$ participates in the training of the $m$-th model, and $\boldsymbol{X}_{m,i} = -\frac{1}{\sqrt{v(1-p)}}$ otherwise, where $v = \mathbb{E}_p[\frac{1}{p(1-p)}]$ and $p$ represents the sampling rate for each training point.

The AME values of the training data are subsequently computed using Lasso regression as follows:

$$\hat{\boldsymbol{\beta}} = \arg\min_{\boldsymbol{\beta} \in \mathbb{R}^N} \left[ ||\boldsymbol{\mathcal{U}} - \boldsymbol{X}\boldsymbol{\beta}||_2^2 + \lambda||\boldsymbol{\beta}||_1 \right], \tag{3}$$

where $\hat{\boldsymbol{\beta}} \in \mathbb{R}^N$ is the optimal linear fit on the $(\boldsymbol{X}, \boldsymbol{\mathcal{U}})$ dataset, which contains the values of all training samples; $\boldsymbol{\mathcal{U}} \in \mathbb{R}^{\mathcal{M}}$ refers to the utility vector derived from $\mathcal{M}$ trained models. Specifically, $\mathcal{U}_m$ denotes the utility of the $m$th model. $\lambda$ is a hyperparameter that governs the strength of regularization. Obviously, Eq. (3) implicitly assumes the Laplace distribution prior (i.e., sparse assumption) for values of samples in the dataset. The advantage of this prior is that the number of sampled subsets, $\mathcal{M}$, can be much smaller than $N$. Since the training time for a single model can be considerable in many tasks, selecting a smaller value for $\mathcal{M}$ can significantly reduce the overall time cost, particularly when $N$ is large for a given dataset.

## 4.2 GLOBAL AND LOCAL CHARACTERISTICS-BASED DATA VALUATION APPROACH

The optimization problem utilized by AME can be reformulated as follows:

$$\hat{\boldsymbol{\beta}} = \arg \min_{\boldsymbol{\beta} \in \mathbb{R}^N} \left[ ||\boldsymbol{\mathcal{U}} - \boldsymbol{X}\boldsymbol{\beta}||_2^2 + \lambda \mathcal{R}_g(\boldsymbol{\beta}) \right], \tag{4}$$

where $\mathcal{R}_g(\cdot)$ denotes a regularizer that incorporates the global statistical prior for the data values. Based on our empirical analysis, the value distribution is more accurately modeled by a Gaussian distribution rather than a Laplace distribution. Therefore, the regularization term $\mathcal{R}_g(\boldsymbol{\beta})$ should be set to $\mathcal{R}_g(\boldsymbol{\beta}) = ||\boldsymbol{\beta}||_2^2$, thereby transforming the optimization problem into a Ridge regression.

Meanwhile, based on our findings regarding the local statistical characteristics, which indicate that the similarities in values between adjacent data points within the same category are substantial, while those between adjacent samples from different classes are minimal, we propose a carefully designed regularization term to refine the data values: $\mathcal{R}_l(\boldsymbol{\beta}) = \sum_{\boldsymbol{x}_i \in \mathcal{D}} \sum_{\boldsymbol{x}_j \in \mathcal{N}_k(\boldsymbol{x}_i)} \mathcal{S}_{i,j}(\beta_i - \beta_j)^2$, where $\beta_i$ and $\beta_j$ denote the values associated with $\boldsymbol{x}_i$ and $\boldsymbol{x}_j$, respectively. $\mathcal{N}_k(\boldsymbol{x}_i)$ denotes the $k$-nearest neighborhood of the sample $\boldsymbol{x}_i$. $\mathcal{S}_{i,j}$ is designed to capture the similarity between the values of samples $\boldsymbol{x}_i$ and $\boldsymbol{x}_j$, with consideration given to both their labels and feature similarities. Specifically, for samples from the same category, the smaller the distance between them, the smaller the difference in their values should be. In contrast, for samples from different categories, the smaller the distance between them, the larger the difference in their values should be. Therefore, the following similarity metric $\mathcal{S}_{i,j}$ is defined: $\mathcal{S}_{i,j} = \cos(\boldsymbol{x}_i, \boldsymbol{x}_j) \cdot [2\mathcal{I}(y_i = y_j) - 1]$[3]. The cosine similarity $\cos(\boldsymbol{x}_i, \boldsymbol{x}_j)$ is computed as $\cos(\boldsymbol{x}_i, \boldsymbol{x}_j) = \frac{\boldsymbol{x}_i \cdot \boldsymbol{x}_j}{|\boldsymbol{x}_i||\boldsymbol{x}_j|}$. Moreover, $\mathcal{I}(\cdot)$ represents a indicator function. If $y_i = y_j$, then $\mathcal{S}_{i,j} = \cos(\boldsymbol{x}_i, \boldsymbol{x}_j)$; if $y_i \neq y_j$, then $\mathcal{S}_{i,j} = -\cos(\boldsymbol{x}_i, \boldsymbol{x}_j)$.

Consequently, our proposed **G**lobal and **LO**cal **C**haracteristics-based data valuation approach, termed GLOC, calculates data values by solving the following optimization problem:

$$\hat{\boldsymbol{\beta}} = \arg \min_{\boldsymbol{\beta} \in \mathbb{R}^N} \left[ ||\boldsymbol{\mathcal{U}} - \boldsymbol{X}\boldsymbol{\beta}||_2^2 + \lambda_1 \mathcal{R}_g(\boldsymbol{\beta}) + \lambda_2 \mathcal{R}_l(\boldsymbol{\beta}) \right], \tag{5}$$

where the two regularizers, $\mathcal{R}_g(\cdot)$ and $\mathcal{R}_l(\cdot)$, are defined as follows:

$$\mathcal{R}_g(\boldsymbol{\beta}) = ||\boldsymbol{\beta}||_2^2,$$
$$\mathcal{R}_l(\boldsymbol{\beta}) = \sum_{\boldsymbol{x}_i \in \mathcal{D}} \sum_{\boldsymbol{x}_j \in \mathcal{N}_k(\boldsymbol{x}_i)} \mathcal{S}_{i,j}(\beta_i - \beta_j)^2. \tag{6}$$

The hyperparameters $\lambda_1$ and $\lambda_2$ control the strengths of the global and local regularizers, respectively. The algorithm for GLOC is provided in Algorithm 1 of the Appendix.

## 4.3 GLOBAL AND LOCAL CHARACTERISTICS-BASED DYNAMIC DATA VALUATION APPROACH

We further propose two dynamic data valuation methods (termed IncGLOC and DecGLOC) based on the identified global and local distribution characteristics, specifically designed for scenarios involving the addition of new data and the removal of existing data.

Here, we focus on incremental data valuation, while the optimization for decremental data valuation follows a similar approach, which is detailed in Appendix A.2. Let the original dataset be $\mathcal{D}$, containing $N$ samples, and the new data to be added be $\mathcal{D}'$, with $N'$ samples. The augmented dataset is denoted as $\hat{\mathcal{D}} = \mathcal{D} \cup \mathcal{D}'$, and let $\boldsymbol{\beta}^{cur}$ represent the original data values in $\mathcal{D}$.

Unlike the only existing work on dynamic data valuation (Zhang et al., 2023), which depends on recalculating Shapley values, this study explores an alternative approach that circumvents their re-estimation to enhance efficiency. Specifically, we aim to explore whether it is possible to infer the values of all data in $\hat{\mathcal{D}}$ based solely on the dataset $\hat{\mathcal{D}}$ and the original data values, $\boldsymbol{\beta}^{cur}$.

As empirically analyzed in Section 3, the changes in value should align with the following insights:

- After incorporating $\mathcal{D}'$ into $\mathcal{D}$, the values of the samples in $\mathcal{D}$ will be adjusted. However, the changes in data values before and after the inclusion of new data are anticipated to remain within a limited range.

---

[3]A performance comparison across different similarity metrics is presented in Appendix A.12.

- The global value distribution of samples in $\hat{\mathcal{D}}$ is expected to follow a Gaussian distribution.
- The values of all data in $\hat{\mathcal{D}}$ should follow the principle of neighborhood consistency, whereby adjacent samples from the same category exhibit similar values, while those from different categories display distinct value differences.

Based on the aforementioned observations, we formulate the following optimization problem to determine the values of the samples in the expanded dataset $\hat{\mathcal{D}}$:

$$\min_{\boldsymbol{\beta}} \sum_{\boldsymbol{x}_i \in \hat{\mathcal{D}}} \sum_{\boldsymbol{x}_j \in \mathcal{N}_k(\boldsymbol{x}_i)} \mathcal{S}_{i,j}(\beta_i - \beta_j)^2 + \eta_1 ||\boldsymbol{\beta}||_2^2, \tag{7}$$
$$\text{s.t.}, |\beta_i^{cur} - \beta_i| \le \epsilon_i, \forall \boldsymbol{x}_i \in \mathcal{D},$$

where $\epsilon_i$ represents the upper bound on the permissible variation in the value of $\boldsymbol{x}_i$. Its value depends on both the variation in the dataset, quantified by the ratio $|\hat{\mathcal{D}}|/|\mathcal{D}|$, and the neighborhood of $\boldsymbol{x}_i$. In general, as the dataset variation increases, $\epsilon_i$ also increases. Similarly, larger variation within the neighborhood of a data point leads to a greater value difference, thereby increasing $\epsilon_i$. Based on these insights, we propose a heuristic definition for $\epsilon_i$, which has been empirically validated for effectiveness in our experiments: $\epsilon_i = \frac{|\hat{\mathcal{D}}|}{|\mathcal{D}|} [1 + r_{\mathcal{N}_k}(\boldsymbol{x}_i)] \epsilon_0$, where $r_{\mathcal{N}_k}(\boldsymbol{x}_i)$ represents the variation ratio within the $k$-nearest neighborhood of $\boldsymbol{x}_i$. Specifically, if all $k$-nearest neighbors undergo changes, then $r_{\mathcal{N}_k}(\boldsymbol{x}_i) = 1$; conversely, if all of its $k$-nearest neighbors remain unchanged, then $r_{\mathcal{N}_k}(\boldsymbol{x}_i) = 0$. Additionally, $\epsilon_0$ is a constant that remains uniform across all samples.

To facilitate solving Eq. (7), we reformulate it as the following unconstrained optimization problem:

$$\min_{\boldsymbol{\beta}} [\sum_{\boldsymbol{x}_i \in \hat{\mathcal{D}}} \sum_{\boldsymbol{x}_j \in \mathcal{N}_k(\boldsymbol{x}_i)} \mathcal{S}_{i,j}(\beta_i - \beta_j)^2 + \eta_1 ||\boldsymbol{\beta}||_2^2 + \eta_2 \sum_{\boldsymbol{x}_i \in \mathcal{D}} \frac{\epsilon_i}{\bar{\epsilon}} (\beta_i^{cur} - \beta_i)^2], \tag{8}$$

where $\eta_1$ and $\eta_2$ control the relative importance of the three objectives. To expedite the optimization of Eq. (8), the initial values for the data in $\mathcal{D}'$ can be assigned as follows:

$$\beta_i = \frac{\sum_{\boldsymbol{x}_j \in \mathcal{N}_k(\boldsymbol{x}_i) \& \boldsymbol{x}_j \in \mathcal{D}} \mathcal{S}_{i,j} \beta_j^{cur}}{\sum_{\boldsymbol{x}_j \in \mathcal{N}_k(\boldsymbol{x}_i) \& \boldsymbol{x}_j \in \mathcal{D}} \mathcal{S}_{i,j}}. \tag{9}$$

This initialization is actually a weighted average of the original values of the samples in the neighborhood of $\boldsymbol{x}_i$, with the weights determined by their similarities.

The decremental data valuation approach follows a similar pipeline and is provided in detail in Appendix A.2 due to space limitations. In contrast to existing approaches that require re-computation of Shapley values under dynamic data scenarios, our method directly infers the updated values by leveraging characteristics of value distributions and patterns of value variation, thereby significantly enhancing computational efficiency.

### 4.4 ADAPTATION TO ALTERNATIVE DATA VALUATION APPROACHES

This study introduces a distribution-aware perspective for data valuation. The proposed regularizers can be easily integrated with most existing valuation methods, except for AME. Specifically, the regularization terms related to value distributions can be employed to optimize data values, either alongside the original method or afterward. The first scenario, which combines our regularizers with other methods, has been demonstrated using AME. In the second scenario, the regularizers are directly utilized as optimization objectives to refine the obtained data values. This approach has been demonstrated to enhance the effectiveness of other valuation methods, as detailed in Appendix A.8.

## 5 EXPERIMENTS

Our experimental investigations are divided into three main components[4]. First, we evaluate the performance of GLOC in Shapley value estimation. Second, we examine two downstream valuation tasks: value-based point addition and removal, as well as mislabeled data detection, to validate the effectiveness of GLOC in identifying valuable and poisoned samples. Finally, we assess the performance of our proposed dynamic data valuation methods, IncGLOC and DecGLOC, in Shapley value estimation under incremental and decremental data valuations.

[4]Our code is available in the submitted supplementary materials.

| Dataset | Electricity | MiniBooNE | CIFAR10 | BBC | Fried | 2Dplanes |
|---------|-------------|-----------|---------|-----|-------|----------|
| Ratio   | 50:1        | 8:1       | 96:1    | 6:1 | 82:1  | 105:1    |
| Dataset | Pol         | Covertype | Nomao   | Law | Creditcard | Jannis |
| Ratio   | 7:1         | 113:1     | 44:1    | 18:1 | 54:1 | 206:1    |

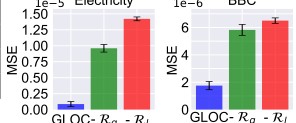

Table 1: Comparison of Shapley value estimation. Ratios of MSEs between AME and GLOC (simplified to the simplest integer ratio) are reported. The MSEs for GLOC are consistently smaller than those for AME, highlighting its superiority in Shapley value estimation.

Figure 3: Ablation studies to the two regularization terms: $\mathcal{R}_g$ and $\mathcal{R}_l$.

**Datasets and Compared Baselines.** Building on prior research (Jiang et al., 2023; Kwon & Zou, 2023), we evaluate our methods on twelve classification datasets spanning tabular, text, and image domains, including Electricity (Gama et al., 2004), MiniBooNE (Roe et al., 2005), CIFAR10 (Krizhevsky et al., 2009), BBC (Greene & Cunningham, 2006), Fried (Breiman, 1996), 2Dplanes, Pol, Covertype, Nomao (Candillier & Lemaire, 2012), Law, Creditcard (Dal Pozzolo et al., 2015), and Jannis. Dataset details are summarized in Table 3 in the Appendix. We benchmark against representative data valuation methods, including AME (Lin et al., 2022), LOO (Jiang et al., 2023), Influence Function (Koh & Liang, 2017), DVRL (Yoon et al., 2020), Data Shapley (Ghorbani & Zou, 2019), KNN Shapley (Jia et al., 2019), Volume-based Shapley (Xu et al., 2021), Beta Shapley (Kwon & Zou, 2022), Data Banzhaf (Wang & Jia, 2023), LAVA (Just et al., 2023), CS-Shapley (Schoch et al., 2022), Du-Shapley (Garrido Lucero et al., 2025), and Data-OOB (Kwon & Zou, 2023) (Appendix A.4). Further experimental settings are provided in Appendix A.5.

**Experiments on Shapley Value Estimation.** This section evaluates the effectiveness of GLOC and AME in estimating Shapley values. Given the ground-truth Shapley values ($SV$) and estimates ($\boldsymbol{\beta}$) from AME and GLOC, the MSE between estimated and true values is defined as $MSE(SV, \boldsymbol{\beta}) = \frac{1}{|\mathcal{D}|} \sum_{i=1}^{|\mathcal{D}|} (SV_i - \beta_i)^2$. Table 1 reports the ratio of MSEs between AME and GLOC in Shapley value estimation, demonstrating that GLOC consistently achieves lower MSEs across various datasets. These results manifest that GLOC provides a closer approximation to the true Shapley values compared to AME, making it a more accurate and effective approach for assessing the contribution of training samples. Additionally, ablation studies are conducted to evaluate the effectiveness of the proposed global ($\mathcal{R}_g$) and local ($\mathcal{R}_l$) regularizers. As shown in Fig. 3, GLOC achieves optimal performance when incorporating two regularizers, highlighting the importance of leveraging both global and local value distributions in data valuation. Further ablation studies regarding Mean Absolute Error (MAE) and Spearman correlation metrics are presented in Appendix A.14.

**Experiments on Value-based Point Addition and Removal.** To assess GLOC's ability to differentiate valuable from harmful samples, we perform point addition and removal experiments (Ghorbani & Zou, 2019; Kwon & Zou, 2023). For point removal, data are sequentially eliminated from the training set by descending value, with a logistic regression retrained after each step and evaluated on a holdout set. Ideally, removing the most informative samples first could result in a degradation of model performance. Conversely, for point addition, data points are introduced in ascending order of their values. Similar to the removal process, model accuracy is expected to remain low initially, as detrimental samples are added first. All experiments are conducted on a perturbed dataset with 20% label noise, with the holdout test set containing 3K samples. Figs. 4(a) and (b) compare the performance of different valuation methods in the context of data removal. GLOC consistently exhibits the most significant decline in performance, highlighting its effectiveness in identifying high-quality samples. Similarly, from Figs. 4(c) and (d), GLOC demonstrates the worst performance, underscoring its ability to detect poisoned data. Additional results are presented in Appendix A.9.

**Experiments on Mislabeled Data Detection.** Mislabeled samples often degrade model performance (Xiong et al., 2006), making it important to assign them low values. Previous studies showed

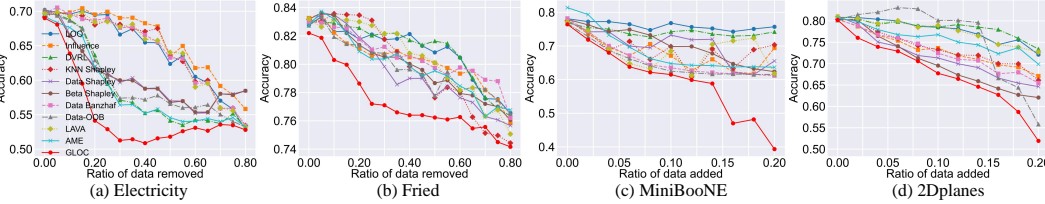

Figure 4: Accuracy variation across different ratios of removed and added data points. We prioritize removing data points with larger values and adding those with smaller values.

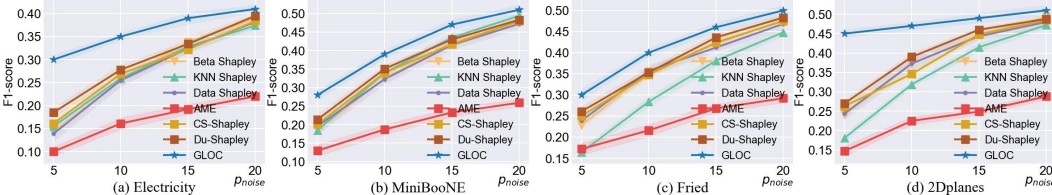

Figure 5: F1-scores for noise detection at varying noise ratios across four datasets. GLOC consistently outperforms the compared baselines in detection performance across various noise levels.

| Manner | Add | | | | Remove | | | |
|---|---|---|---|---|---|---|---|---|
| Dataset | Electricity | MiniBooNE | CIFAR10 | Fried | Electricity | MiniBooNE | CIFAR10 | Fried |
| MC | 5.76e-5 | 7.95e-5 | 4.65e-5 | 2.57e-5 | 7.63e-6 | 5.06e-6 | 4.89e-5 | 1.21e-5 |
| TMC | 8.75e-4 | 1.25e-4 | 4.89e-4 | 1.23e-5 | 4.43e-5 | 6.08e-5 | 5.77e-4 | 3.42e-4 |
| Delta | 7.76e-6 | 4.78e-6 | 8.91e-6 | 4.88e-6 | 3.89e-5 | 3.58e-5 | 2.78e-5 | 1.29e-5 |
| KNN | 3.88e-5 | 5.67e-6 | 2.45e-5 | 5.34e-5 | 7.65e-6 | 6.93e-6 | 6.79e-6 | 4.32e-5 |
| KNN+ | 3.45e-5 | 4.56e-5 | 5.24e-5 | 6.45e-6 | 2.48e-6 | 5.67e-6 | 3.74e-5 | 4.56e-5 |
| Ours | **1.73e-6** | **1.99e-6** | **3.29e-6** | **2.17e-6** | **0.95e-6** | **2.00e-6** | **2.55e-6** | **2.27e-6** |

Table 2: MSEs for data addition and removal, with the best and second-best results highlighted in bold and underline, respectively. Our methods (IncGLOC and DecGLOC) achieve the lowest MSEs, demonstrating their superior approximation to Shapley values.

that AME performs poorly in detecting mislabeled data. This section compares the detection capabilities of GLOC with several Shapley value-based valuation approaches. We randomly select $p_{noise}\%$ of the entire dataset and alter their labels to one of the other classes. Four different noise levels are considered: $p_{noise} \in \{5, 10, 15, 20\}$. Using K-means (MacQueen, 1967), we cluster samples based on their values into two groups. Points in the cluster with the lowest mean values are identified as mislabeled. F1-scores are computed by comparing predictions with true labels. Fig. 5 shows that GLOC consistently outperforms other methods across four datasets and varying noise levels.

**Experiments on Dynamic Data Valuation.** This section evaluates the proposed dynamic data valuation methods, IncGLOC and DecGLOC, under sample addition and removal scenarios. The average MSE is also used to assess the effectiveness of different methods in estimating Shapley values. In accordance with the only existing research on dynamic data valuation (Zhang et al., 2023), the compared methods include Monte Carlo Shapley (MC), Delta-based algorithm (Delta), KNN-based algorithm (KNN), KNN+-based algorithm (KNN+), which are proposed by (Zhang et al., 2023), and Truncated Monte Carlo Shapley (TMC) (Ghorbani & Zou, 2019). For fair comparison, all methods except MC and TMC, which compute Shapley values from scratch, are initialized with MC-estimated data values. Table 2 presents the results for adding or removing a single data point, while results for multiple data points and other ground-truth settings are provided in Appendix A.11. IncGLOC and DecGLOC consistently achieve the lowest MSEs across various datasets, demonstrating their effectiveness in Shapley value estimation under dynamic data conditions.

Additionally, we compare the computational complexity of various data valuation methods to assess the efficiency of our proposed approaches. The results are provided in Fig. 9 of the Appendix. Methods such as KNN, KNN+, and our approaches derive updated data values from current values without recalculating the Shapley values, resulting in low time consumption. In contrast, methods that require re-estimating the Shapley values, such as MC and TMC, entail significant computational overhead for dynamic data valuation, even when adding or deleting a single data point.

# 6 CONCLUSION

This study proposes the integration of global and local statistical information of data values into the data valuation process, a perspective that has often been overlooked by previous approaches. By examining the characteristics of value distributions, we introduce a new data valuation method that incorporates these distribution characteristics as regularization terms. Furthermore, we present two dynamic data valuation algorithms designed for incremental and decremental data valuation, respectively. These algorithms compute data values based solely on the original and updated datasets, alongside the original data values, without requiring additional Shapley value estimation steps, thus ensuring computational efficiency. Extensive experiments across various tasks, such as Shapley value estimation, point addition and removal, mislabeled data detection, and dynamic data valuation, demonstrate the significant effectiveness and efficiency of the proposed methodologies.

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

## A  APPENDIX

### A.1  CALCULATION PROCEDURE FOR GLOC

The complete algorithm for our proposed GLOC approach is outlined in Algorithm 1.

### A.2  ALGORITHMS FOR DYNAMIC DATA VALUATION

The derivation of our proposed IncGLOC for incremental data valuation is presented in the main text, with the corresponding algorithm provided in Algorithm 2. In the following, we outline the derivation of our method for decremental data valuation.

In the context of decremental data valuation, a subset $\mathcal{D}'$ containing $N'$ samples is removed from the existing dataset $\mathcal{D}$, which contains $N$ training samples. The resulting dataset after the removal is denoted as $\widehat{\mathcal{D}} = \mathcal{D} - \mathcal{D}'$. Let $\boldsymbol{\beta}^{cur}$ represent the current values of the samples in dataset $\mathcal{D}$. The core question we address is whether we can infer the values of data points in $\widehat{\mathcal{D}}$ using only the dataset $\widehat{\mathcal{D}}$ and the original data values $\boldsymbol{\beta}^{cur}$.

---

**Algorithm 1:** Algorithm of GLOC.

---

**Input:** Training data $\mathcal{D} = \{(\boldsymbol{x}_i, y_i)\}_{i=1}^N$, number of sampled subsets $\mathcal{M}$, probability distribution $\mathcal{P} = \text{Uniform}\{p_1, p_2, \cdots, p_{\mathcal{J}}\}$, regularization hyperparameters $\lambda_1$ and $\lambda_2$, neigborhood size $k$, and others.

**Output:** Values $\boldsymbol{\beta}$ for all data points in $\mathcal{D}$.

**1** Initialize $\boldsymbol{X} \leftarrow zeros(\mathcal{M}, N); \boldsymbol{\mathcal{U}} \leftarrow zeros(\mathcal{M})$;

**2** **for** $m \leftarrow 1$ *to* $\mathcal{M}$ **do**

**3** $\quad$ $\mathcal{B}_m \leftarrow \{\}, p \sim \mathcal{P}$;

**4** $\quad$ **for** $i \leftarrow 1$ *to* $N$ **do**

**5** $\quad\quad$ $r \sim \text{Bernoulli}(p)$;

**6** $\quad\quad$ **if** $r = 1$ **then**

**7** $\quad\quad\quad$ $\mathcal{B}_m \leftarrow \mathcal{B}_m + \{(\boldsymbol{x}_i, y_i)\}$;

**8** $\quad\quad$ **end**

**9** $\quad\quad$ $\boldsymbol{X}_{m,i} \leftarrow \frac{r}{p} - \frac{1-r}{1-p}$;

**10** $\quad$ **end**

**11** **end**

**12** Calculate the feature similarity $\mathcal{S}$ between each pair of samples in the dataset $\mathcal{D}$;

**13** **for** $m \leftarrow 1$ *to* $\mathcal{M}$ **do**

**14** $\quad$ Calculate $\mathcal{U}_m$ using the model trained on the $m$th training subset $\mathcal{B}_m$;

**15** **end**

**16** $\hat{\boldsymbol{\beta}} \leftarrow \arg\min\limits_{\boldsymbol{\beta} \in \mathbb{R}^N} \left[ ||\boldsymbol{\mathcal{U}} - \boldsymbol{X}\boldsymbol{\beta}||_2^2 + \lambda_1 \mathcal{R}_g(\boldsymbol{\beta}) + \lambda_2 \mathcal{R}_l(\boldsymbol{\beta}) \right]$, with the regularizers defined in Eq. (6);

---

Similar to the optimization problem formulated for incremental data valuation, we formulate the following optimization problem for decremental data valuation:

$$\min_{\boldsymbol{\beta}} \sum_{\boldsymbol{x}_i \in \widehat{\mathcal{D}}} \sum_{\boldsymbol{x}_j \in \mathcal{N}_k(\boldsymbol{x}_i)} \mathcal{S}_{i,j}(\beta_i - \beta_j)^2 + \eta_1 ||\boldsymbol{\beta}||_2^2,$$

$$\text{s.t., } |\beta_i^{cur} - \beta_i| \leq \epsilon_i, \forall \boldsymbol{x}_i \in \widehat{\mathcal{D}}. \tag{10}$$

The permissible variation bound, $\epsilon_i$ is also determined by the variation within the dataset and the neighborhood of the samples, and is calculated as follows: $\epsilon_i = \frac{|\mathcal{D}|}{|\widehat{\mathcal{D}}|}(1 + r_{\mathcal{N}_k}(\boldsymbol{x}_i))\epsilon_0$. To facilitate solving Eq. (10), it is reformulated as the following unconstrained optimization problem:

$$\min_{\boldsymbol{\beta}} \sum_{\boldsymbol{x}_i \in \widehat{\mathcal{D}}} \sum_{\boldsymbol{x}_j \in \mathcal{N}_k(\boldsymbol{x}_i)} \mathcal{S}_{i,j}(\beta_i - \beta_j)^2 + \eta_1 ||\boldsymbol{\beta}||_2^2 + \eta_2 \sum_{\boldsymbol{x}_i \in \widehat{\mathcal{D}}} \frac{\epsilon_i}{\bar{\epsilon}}(\beta_i^{cur} - \beta_i)^2, \tag{11}$$

where $\eta_1$ and $\eta_2$ are two hyperparameters that control the relative strengths of the three optimization objectives. For simplicity, the procedure for computing data values after removing a set of samples is denoted as DecGLOC. The detailed algorithmic steps of DecGLOC are presented in Algorithm 3.

## A.3 DATASET DESCRIPTION

This section provides a detailed description of the applied datasets. First, we detail the synthetic dataset compiled for analyzing the global and local distributional properties of data values. The simulated dataset, referred to as "Random," is generated by randomly sampling from the following data distribution:

$$y \overset{u.a.r}{\sim} \{-1, +1\}, \quad \boldsymbol{\theta} = [+1, +1]^T \in \mathbb{R}^2,$$

$$\boldsymbol{x} \sim \begin{cases} \mathcal{N}\left(\boldsymbol{\theta}, \sigma_+^2 \boldsymbol{I}\right), & \text{if } y = +1 \\ \mathcal{N}\left(-\boldsymbol{\theta}, \sigma_-^2 \boldsymbol{I}\right), & \text{if } y = -1 \end{cases}, \tag{12}$$

where $\mathcal{N}(\boldsymbol{\theta}, \sigma_+^2 \boldsymbol{I})$ denotes a Gaussian distribution, with the mean $\boldsymbol{\theta}$ and the variance $\sigma_+^2 \boldsymbol{I}$, and $\boldsymbol{I}$ represents an identity matrix. A $K$-factor difference is set between two classes' variances, that is $\sigma_+ : \sigma_- = K : 1$ and $K = 2$. Moreover, $\sigma_- = 1$. The training and test sets each contain 5K sampled data points for both categories. Considering that, for simulated data, assuming $\boldsymbol{x}$ comes from a Gaussian distribution, it is not entirely convincing to claim that the resulting value distribution

---

**Algorithm 2:** Algorithm of IncGLOC.

---

**Input:** $\mathcal{D}$ and $\mathcal{D}'$, original data values $\boldsymbol{\beta}^{cur}$ for instances in $\mathcal{D}$, neighborhood size $k$, hyperparameters $\eta_1$, $\eta_2$, and $\epsilon_0$, and others.

**Output:** Values $\boldsymbol{\beta}$ of all data points in $\hat{\mathcal{D}} = \mathcal{D} \cup \mathcal{D}'$.

**1** Calculate the similarity $\boldsymbol{S}$ for samples in $\hat{\mathcal{D}}$;

**2** Initialize data values $\beta_i$ for $\boldsymbol{x}_i \in \mathcal{D}'$ using Eq. (9);

**3** Calculate the original neighborhood $\mathcal{N}_k^{ori}(\boldsymbol{x}_i)$ for $\boldsymbol{x}_i \in \mathcal{D}$;

**4** Calculate the neighborhood $\mathcal{N}_k(\boldsymbol{x}_i)$ after adding $\mathcal{D}'$ for $\boldsymbol{x}_i \in \hat{\mathcal{D}}$;

**5** $r_{\mathcal{N}_k}(\boldsymbol{x}_i) \leftarrow \frac{|\mathcal{N}_k(\boldsymbol{x}_i) - \mathcal{N}_k^{ori}(\boldsymbol{x}_i)|}{k}$ for $\boldsymbol{x}_i \in \mathcal{D}$;

**6** $\epsilon_i \leftarrow \frac{|\hat{\mathcal{D}}|}{|\mathcal{D}|}(1 + r_{\mathcal{N}_k}(\boldsymbol{x}_i))\epsilon_0$ for $\boldsymbol{x}_i \in \mathcal{D}$;

**7** $\bar{\epsilon} = \frac{1}{|\mathcal{D}|}\sum_{i=1}^{|\mathcal{D}|}\epsilon_i$;

**8** $\hat{\boldsymbol{\beta}} \leftarrow \arg\min_{\boldsymbol{\beta}}\sum_{\boldsymbol{x}_i \in \hat{\mathcal{D}}}\sum_{\boldsymbol{x}_j \in \mathcal{N}_k(\boldsymbol{x}_i)}\mathcal{S}_{i,j}(\beta_i - \beta_j)^2 + \eta_1||\boldsymbol{\beta}||_2^2 + \eta_2\sum_{\boldsymbol{x}_i \in \mathcal{D}}\frac{\epsilon_i}{\bar{\epsilon}}(\beta_i^{cur} - \beta_i)^2$.

---

**Algorithm 3:** Algorithm of DecGLOC.

---

**Input:** $\mathcal{D}$ and $\mathcal{D}'$, original data values $\boldsymbol{\beta}^{cur}$ for instances in $\mathcal{D}$, neighborhood size $k$, hyperparameters $\eta_1$, $\eta_2$, and $\epsilon_0$, and others.

**Output:** Values $\boldsymbol{\beta}$ of all data points in $\widehat{\mathcal{D}} = \mathcal{D} - \mathcal{D}'$.

**1** Calculate the original neighborhood $\mathcal{N}_k^{ori}(\boldsymbol{x}_i)$ for $\boldsymbol{x}_i \in \mathcal{D}$;

**2** Calculate the new neighborhood $\mathcal{N}_k(\boldsymbol{x}_i)$ after deleting $\mathcal{D}'$ for $\boldsymbol{x}_i \in \widehat{\mathcal{D}}$;

**3** Calculate the similarity $\boldsymbol{S}$ for samples in $\widehat{\mathcal{D}}$;

**4** $r_{\mathcal{N}_k}(\boldsymbol{x}_i) \leftarrow \frac{|\mathcal{N}_k(\boldsymbol{x}_i) - \mathcal{N}_k^{ori}(\boldsymbol{x}_i)|}{k}$ for $\boldsymbol{x}_i \in \widehat{\mathcal{D}}$ ;

**5** $\epsilon_i \leftarrow \frac{|\mathcal{D}|}{|\widehat{\mathcal{D}}|}(1 + r_{\mathcal{N}_k}(\boldsymbol{x}_i))\epsilon_0$ for $\boldsymbol{x}_i \in \widehat{\mathcal{D}}$;

**6** $\bar{\epsilon} = \frac{1}{|\widehat{\mathcal{D}}|}\sum_{i=1}^{|\widehat{\mathcal{D}}|}\epsilon_i$;

**7** $\hat{\boldsymbol{\beta}} \leftarrow \arg\min_{\boldsymbol{\beta}}\sum_{\boldsymbol{x}_i \in \widehat{\mathcal{D}}}\sum_{\boldsymbol{x}_j \in \mathcal{N}_k(\boldsymbol{x}_i)}\mathcal{S}_{i,j}(\beta_i - \beta_j)^2 + \eta_1||\boldsymbol{\beta}||_2^2 + \eta_2\sum_{\boldsymbol{x}_i \in \widehat{\mathcal{D}}}\frac{\epsilon_i}{\bar{\epsilon}}(\beta_i^{cur} - \beta_i)^2$.

---

is Gaussian-like. To address this, we also explore another synthetic dataset in which each feature dimension follows an independent Laplace distribution:

$$y \overset{u.a.r}{\sim} \{-1, +1\}, \quad \boldsymbol{\theta} = [+1, +1]^T \in \mathbb{R}^2,$$
$$\boldsymbol{x} \sim \begin{cases} Laplace(\boldsymbol{\theta}, b_+), & \text{if } y = +1 \\ Laplace(-\boldsymbol{\theta}, b_-), & \text{if } y = -1 \end{cases}, \tag{13}$$

where $b_+$ and $b_-$ denote the scale parameters for Classes "+1" and "-1", respectively. We then perform Shapley value estimation and distributional analysis on this dataset. The KStest yields a $p$-value of 0.5816, indicating that the resulting value distribution is consistent with a Gaussian distribution.

Following prior research (Jiang et al., 2023; Kwon & Zou, 2023), we also examine a variety of real-world datasets to analyze the characteristics of value distributions and assess the effectiveness of the proposed data valuation approaches. The applied twelve classification datasets, spanning tabular, text, and image types, include Electricity (Gama et al., 2004), MiniBooNE (Roe et al., 2005), CIFAR-10 (Krizhevsky et al., 2009), BBC (Greene & Cunningham, 2006), Fried (Breiman, 1996), 2Dplanes, Pol, Covertype, Nomao (Candillier & Lemaire, 2012), Law, Creditcard (Dal Pozzolo et al., 2015), and Jannis. Each dataset undergoes standard normalization, ensuring that all features have a zero mean and unit standard deviation. After preprocessing, the data is divided into three subsets: training, validation, and test datasets. Detailed information on these datasets is provided in Table 3.

### A.4 COMPARED BASELINES

A number of advanced data valuation methods from various categories, including marginal contribution-based, gradient-based, importance weight-based, and out-of-bag-based approaches,

| Name | Size | Dimension | # Classes | Source | Minor class proportion |
|------|------|-----------|-----------|--------|------------------------|
| Law | 20800 | 6 | 2 | OpenML-43890 | 0.321 |
| Electricity | 38474 | 6 | 2 | Gama et al. (2004) | 0.5 |
| Fried | 40768 | 10 | 2 | Breiman (1996) | 0.498 |
| 2Dplanes | 40768 | 10 | 2 | OpenML-727 | 0.499 |
| Creditcard | 30000 | 23 | 2 | Dal Pozzolo et al. (2015) | 0.221 |
| Pol | 15000 | 48 | 2 | OpenML-722 | 0.336 |
| MiniBooNE | 72998 | 50 | 2 | Roe et al. (2005) | 0.5 |
| Jannis | 57580 | 54 | 2 | OpenML-43977 | 0.5 |
| Nomao | 34465 | 89 | 2 | Candillier & Lemaire (2012) | 0.285 |
| Covertype | 581012 | 54 | 7 | Scikit-learn | 0.004 |
| BBC | 2225 | 768 | 5 | Greene & Cunningham (2006) | 0.17 |
| CIFAR10 | 50000 | 2048 | 10 | Krizhevsky et al. (2009) | 0.1 |

Table 3: Summary of twelve classification datasets utilized in our experiments.

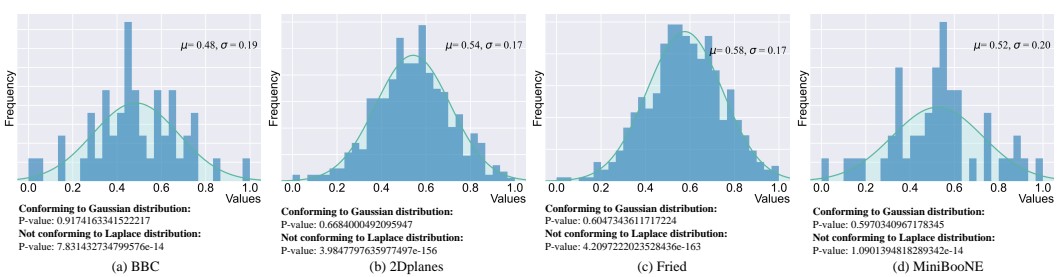

(a) BBC  (b) 2Dplanes  (c) Fried  (d) MiniBooNE

Figure 6: Illustration of the global distributions of data values for four additional datasets: BBC, 2Dplanes, Fried, and MiniBooNE. The results of the KStest hypothesis test (Justel et al., 1997), presented below the figures, indicate that the global value distribution exhibits a closer fit to a Gaussian distribution than to a Laplace distribution.

are compared with our proposed methodologies, including AME (Lin et al., 2022), LOO (Jiang et al., 2023), Influence Function (Koh & Liang, 2017), DVRL (Yoon et al., 2020), Data Shapley (Ghorbani & Zou, 2019), KNN Shapley (Jia et al., 2019), Volume-based Shapley (Xu et al., 2021), Beta Shapley (Kwon & Zou, 2022), Data Banzhaf (Wang & Jia, 2023), LAVA (Just et al., 2023), CS-Shapley Schoch et al. (2022), Du-Shapley Garrido Lucero et al. (2025), and Data-OOB (Kwon & Zou, 2023). A detailed description of these methods is provided below:

- **AME** (Lin et al., 2022): AME quantifies the expected marginal effect of incorporating a sample into various training subsets. When subsets are sampled from the uniform distribution, it equates to the Shapley value.

- **LOO** (Jiang et al., 2023): LOO, belonging to the marginal contribution-based category, measures the utility change when one data point of interest is removed from the entire dataset.

- **Influence Function** (Koh & Liang, 2017): Influence Function is approximated by the difference between two average model performances: one containing a data point of interest in the training procedure and the other not.

- **DVRL** (Yoon et al., 2020): DVRL belongs to the importance weight-based category, involving the utilization of reinforcement learning algorithms to compute data values.

- **Data Shapley** (Ghorbani & Zou, 2019): Data Shapley belongs to the marginal contribution-based category, which takes a simple average of all the marginal contributions.

| Ratio | Original | 5% | 10% | 15% | 20% | 25% | 30% | 35% | 40% |
|-------|----------|-----|------|------|------|------|------|------|------|
| Random | $\mu$=0.56, $\sigma$=0.22 | $\mu$=0.56, $\sigma$=0.21 | $\mu$=0.55, $\sigma$=0.21 | $\mu$=0.53, $\sigma$=0.20 | $\mu$=0.50, $\sigma$=0.21 | $\mu$=0.50, $\sigma$=0.19 | $\mu$=0.47, $\sigma$=0.25 | $\mu$=0.44, $\sigma$=0.26 | $\mu$=0.43, $\sigma$=0.25 |
| Electricity | $\mu$=0.44, $\sigma$=0.15 | $\mu$=0.46, $\sigma$=0.15 | $\mu$=0.49, $\sigma$=0.16 | $\mu$=0.47, $\sigma$=0.17 | $\mu$=0.41, $\sigma$=0.18 | $\mu$=0.39, $\sigma$=0.18 | $\mu$=0.52, $\sigma$=0.24 | $\mu$=0.54, $\sigma$=0.22 | $\mu$=0.57, $\sigma$=0.23 |

Table 4: Variation in data value distribution under different ratios of sample addition and removal.

| Dataset | Pol | Jannis | Law | Covertype | Nomao | Creditcard |
|---|---|---|---|---|---|---|
| KNN Shapley | 0.28 ± 0.003 | 0.25 ± 0.004 | 0.45 ± 0.014 | 0.51 ± 0.021 | 0.47 ± 0.013 | 0.43 ± 0.004 |
| KNN Shapley[†] | **0.73** ± 0.007 | **0.33** ± 0.006 | **0.96** ± 0.011 | **0.55** ± 0.016 | **0.70** ± 0.012 | **0.50** ± 0.006 |
| Data Shapley | 0.50 ± 0.011 | 0.23 ± 0.003 | 0.94 ± 0.003 | 0.37 ± 0.004 | 0.65 ± 0.005 | 0.36 ± 0.006 |
| Data Shapley[†] | **0.77** ± 0.010 | **0.31** ± 0.005 | **0.97** ± 0.008 | **0.51** ± 0.006 | **0.72** ± 0.008 | **0.48** ± 0.008 |
| Beta Shapley | 0.46 ± 0.010 | 0.24 ± 0.003 | 0.94 ± 0.003 | 0.41 ± 0.003 | 0.66 ± 0.005 | 0.43 ± 0.005 |
| Beta Shapley[†] | **0.75** ± 0.009 | **0.30** ± 0.008 | **0.97** ± 0.007 | **0.54** ± 0.005 | **0.74** ± 0.007 | **0.49** ± 0.007 |

Table 5: Results for data values computed using baseline valuation methods, further refined by our proposed regularization terms in noise detection tasks (denoted using "[†]"). The reported values represent the mean and standard error across five independent experiments. The regularization terms regarding value distributions can enhance the accuracy of the obtained data values, further improving the overall detection performance.

- **KNN Shapley** (Jia et al., 2019): KNN Shapley is also founded on the Shapley value but distinguishes itself through the utilization of a utility tailored to $k$-nearest neighbors.

- **Volume-based Shapley** (Xu et al., 2021): The idea of the Volume-based Shapley is to utilize the same Shapley value function as Data Shapley, but it is characterized by using the volume of input data for a utility function.

- **Beta Shapley** (Kwon & Zou, 2022): Beta Shapley has a form of a weighted mean of the marginal contributions, which generalizes Data Shapley by relaxing the efficiency axiom in the Shapley value.

- **Data Banzhaf** (Wang & Jia, 2023): Data Banzhaf, also belonging to the marginal contribution-based category, is founded on the Banzhaf value.

- **LAVA** (Just et al., 2023): LAVA is proposed to measure how fast the optimal transport cost between a training dataset and a validation dataset changes when a training data point of interest is more weighted.

- **CS-Shapley** Schoch et al. (2022): CS-Shapley extends the standard Shapley value by introducing a novel value function that explicitly distinguishes between the in-class and out-of-class contributions of training instances.

- **Du-Shapley** Garrido Lucero et al. (2025): Du-Shapley is formulated as an expectation with respect to a discrete uniform distribution defined over a support of manageable size.

- **Data-OOB** (Kwon & Zou, 2023): Data-OOB is a distinctive data valuation algorithm, which uses the out-of-bag estimate to describe the quality of data.

Additionally, in line with the only study exploring dynamic data valuation by Zhang et al. (2023), the methods compared with our proposed dynamic data valuation approaches, IncGLOC and DecGLOC, include Monte Carlo Shapley (MC), Delta-based algorithm (Delta), KNN-based algorithm (KNN), KNN+-based algorithm (KNN+), which are proposed by (Zhang et al., 2023), and Truncated Monte Carlo Shapley (TMC) (Ghorbani & Zou, 2019). The details of these methods are provided as follows:

- **MC** (Zhang et al., 2023): The MC simulation gives an unbiased estimation of the exact Shapley value. The number of permutations controls the trade-off between approximation error and time cost. A larger number of samples brings a more accurate Shapley value at the expense of more running time.

- **Delta** (Zhang et al., 2023): To further enhance efficiency, Delta represents the difference of Shapley value with the differential marginal contribution, whose absolute value is smaller than the marginal contribution.

- **KNN** (Zhang et al., 2023): This approach is inspired by the observation that data points with similar features tend to have a similar performance on the machine learning models, which results in similar utility functions and similar Shapley value.

- **KNN+** (Zhang et al., 2023): This method learns a regression function for the changes of Shapley values based on their similarity to the new data point, and uses this function to derive the updated Shapley values of the original data points.

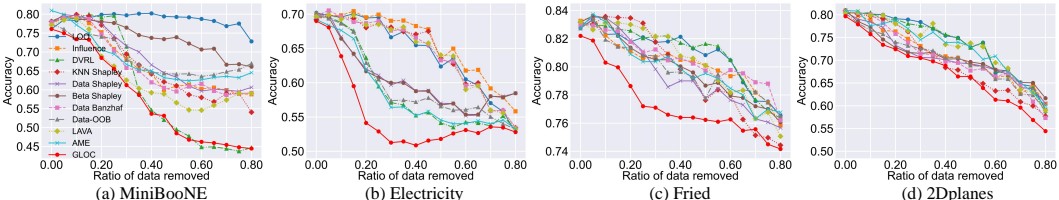

Figure 7: Variation in accuracy across different ratios of removed instances. Data points with the highest values are removed first. GLOC exhibits the lowest accuracy, confirming its effectiveness in identifying influential data points.

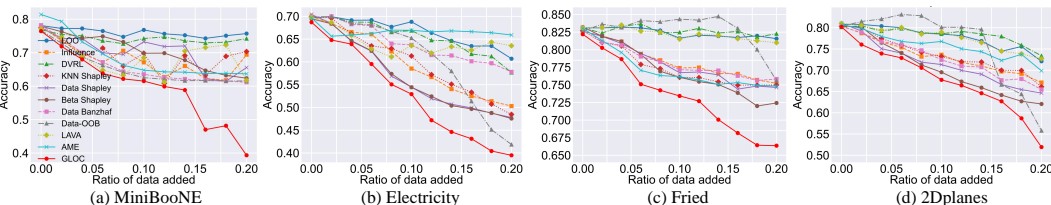

Figure 8: Variation in accuracy across different ratios of added instances. Data points with the lowest values are added first. When only low-value samples are introduced, GLOC exhibits the worst performance, highlighting its ability to identify poisoned samples.

- **TMC** (Ghorbani & Zou, 2019): Instead of scanning over all data sources in the sampled permutation, TMC truncates the computations once the marginal contributions become small and approximates the marginal contribution of the following elements with zero.

## A.5 EXPERIMENTAL CONFIGURATION

The hyperparameters for the AME approach follow the settings outlined in the original paper (Lin et al., 2022). Specifically, the regularization parameter is selected using LassoCV from the Sklearn library. The number of sampled subsets is set to 500, and the data sampling distribution is defined as $\mathcal{P} = \text{Uniform}\{0.2, 0.4, 0.6, 0.8\}$. Moreover, the configurations for the other compared baselines are consistent with those specified in their respective original papers. To assess the effectiveness of the proposed valuation approaches, we utilize the MSE to quantify the difference between the computed data values and the true Shapley values, where the ground-truth Shapley values are calculated using AME based on a large number of sampled subsets, denoted as $\mathcal{M}$, which is equal to the training size of each dataset.

The hyperparameters associated with the regularization terms for our proposed approaches—specifically, $\lambda_1$ and $\lambda_2$ for GLOC, and $\eta_1$ and $\eta_2$ for IncGLOC and DecGLOC—are selected through a standard empirical procedure. This procedure involves performing five-fold cross-validation (CV) and choosing the values that minimize the CV error. The candidate values for $\lambda_1$ and $\lambda_2$ are {1e-2, 1e-3, 1e-4}, for $\eta_1$, the candidate values are {1e-1, 1e-2, 1e-3}, and for $\eta_2$, the candidate values are {1, 5, 10}. The value of $\epsilon_0$ is set to 1, and the neighborhood size parameter, $k$, is set to 5. The base prediction model employed is logistic regression.

For natural language and image datasets, we use pretrained DistilBERT (Sanh et al., 2019) and ResNet50 (He et al., 2016) models to extract embeddings. The sample sizes for the training and validation datasets are set to 1K and 100, respectively. The test dataset size is fixed at 3K for all datasets, except for the text datasets, where it is set to 500. All experiments are conducted on a machine equipped with a single NVIDIA RTX 4090 GPU and 128 GB of RAM.

| Dataset | Method | 0.05 | 0.1 | 0.15 | 0.2 |
|---------|--------|------|-----|------|-----|
| MiniBooNE | AME | 0.54 | 0.62 | 0.69 | 0.74 |
| | GLOC | **0.62** | **0.71** | **0.79** | **0.83** |
| 2Dplanes | AME | 0.57 | 0.63 | 0.70 | 0.75 |
| | GLOC | **0.64** | **0.71** | **0.80** | **0.84** |

Table 6: Comparison of model performance between GLOC and AME under the removal of low-value samples for MiniBooNE and 2Dplanes datasets.

| Dataset | Method | 0.2 | 0.4 | 0.6 | 0.8 |
|---------|--------|-----|-----|-----|-----|
| Electricity | AME | 0.59 | 0.64 | 0.67 | 0.69 |
| | GLOC | **0.61** | **0.67** | **0.71** | **0.73** |
| Fried | AME | 0.70 | 0.72 | 0.76 | 0.79 |
| | GLOC | **0.77** | **0.80** | **0.84** | **0.86** |

Table 7: Comparison of model performance between GLOC and AME under the addition of high-value samples for Electricity and Fried datasets.

## A.6    More Experiments for Empirical Investigation

We present additional results from our empirical investigation into the global and local distributions of data values. Fig. 6 illustrates the global distribution of data values for three additional datasets. As observed, compared to the Laplace distribution, the global distribution of data values more closely resembles a Gaussian distribution. Therefore, we adopt the Gaussian distribution as the prior for modeling the global distribution of data values within a dataset.

## A.7    Further Analysis of Data Value Variation under Addition and Removal

Most existing dynamic data valuation studies consider only minimal modifications to the dataset, often limited to the addition or removal of one or two samples (Zhang et al., 2023). Under such small-scale changes, it is expected that the data value distribution remains largely unaffected. Our experiments (shown in Fig. 2) demonstrate that when removing or adding 10% of the dataset, the change in the distribution of data values remains small.

Nonetheless, this section presents a more comprehensive analysis across a wider range of modification ratios, which would provide deeper insight into the influence of dataset changes. To this end, we systematically increased the ratio of modified data from 5% to 40% in increments of 5%. The results presented in Table 4 yield the following findings:

- When the dataset undergoes modifications of up to 15%, the deviation in the distribution of data valuations remains minimal, staying within $\pm 0.05$ on a normalized scale.
- Once the modification exceeds 30%, the distribution of data valuations begins to exhibit more pronounced shifts.

Given that dynamic data valuation methods are primarily designed for scenarios involving relatively minor dataset modifications, our approach, which updates data values based on both the modified

| Dataset | Pol | Jannis | Law | Covertype | Nomao | Creditcard |
|---------|-----|--------|-----|-----------|-------|------------|
| AME | 0.09 ± 0.009 | 0.09 ± 0.012 | 0.10 ± 0.009 | 0.12 ± 0.011 | 0.08 ± 0.009 | 0.09 ± 0.011 |
| KNN Shapley | 0.28 ± 0.003 | 0.25 ± 0.004 | 0.45 ± 0.014 | 0.51 ± 0.021 | 0.47 ± 0.013 | 0.43 ± 0.004 |
| Data Shapley | 0.50 ± 0.011 | 0.23 ± 0.003 | 0.94 ± 0.003 | 0.37 ± 0.004 | 0.65 ± 0.005 | 0.36 ± 0.006 |
| Beta Shapley | 0.46 ± 0.010 | 0.24 ± 0.003 | 0.94 ± 0.003 | 0.41 ± 0.003 | 0.66 ± 0.005 | 0.43 ± 0.005 |
| CS-Shapley | 0.56 ± 0.008 | 0.25 ± 0.006 | 0.92 ± 0.007 | 0.51 ± 0.009 | 0.65 ± 0.015 | 0.41 ± 0.007 |
| Du-Shapley | 0.57 ± 0.009 | 0.24 ± 0.004 | 0.93 ± 0.006 | 0.50 ± 0.007 | 0.66 ± 0.012 | 0.43 ± 0.005 |
| GLOC | **0.66** ± 0.009 | **0.30** ± 0.007 | **0.96** ± 0.008 | **0.53** ± 0.011 | **0.68** ± 0.006 | **0.46** ± 0.005 |

Table 8: Comparison of F1-scores for the mislabeled data detection tasks. GLOC demonstrates competitive performance compared to all other evaluated approaches.

| Manner | Add | | | | Remove | | | |
|---|---|---|---|---|---|---|---|---|
| Dataset | Electricity | MiniBooNE | CIFAR10 | Fried | Electricity | MiniBooNE | CIFAR10 | Fried |
| MC | 6.32e-6 | 5.12e-5 | 3.51e-5 | 3.68e-5 | 5.67e-6 | 5.81e-5 | 1.85e-5 | 4.47e-5 |
| TMC | 4.92e-5 | 5.48e-4 | 3.24e-4 | 1.21e-4 | 6.73e-5 | 3.21e-4 | 8.91e-5 | 2.67e-5 |
| Delta | 9.67e-7 | 3.24e-5 | 6.77e-6 | 3.87e-5 | 4.36e-6 | 5.43e-5 | 6.44e-6 | 4.55e-5 |
| KNN | 8.98e-6 | 1.29e-5 | 1.89e-5 | 4.21e-5 | 5.03e-6 | 7.85e-6 | 5.62e-5 | 8.97e-6 |
| KNN+ | 4.67e-6 | 4.65e-6 | 4.78e-5 | 9.56e-6 | 2.56e-6 | 3.98e-6 | 3.45e-5 | 3.99e-5 |
| Ours | **2.36e-7** | **2.67e-6** | **3.52e-6** | **2.04e-6** | **1.34e-6** | **2.86e-6** | **2.59e-6** | **2.01e-6** |

Table 9: Comparison of MSEs for the addition and removal of two data points. The data values estimated by our proposed approaches (i.e., IncGLOC and DecGLOC) exhibit the lowest MSEs, thereby demonstrating a closer approximation to the Shapley values.

| Manner | Add | | | | Remove | | | |
|---|---|---|---|---|---|---|---|---|
| Dataset | Electricity | MiniBooNE | CIFAR10 | Fried | Electricity | MiniBooNE | CIFAR10 | Fried |
| MC | 8.78e-6 | 9.43e-6 | 2.67e-5 | 9.62e-6 | 5.45e-6 | 3.44e-6 | 8.72e-6 | 6.47e-6 |
| TMC | 7.87e-5 | 5.67e-5 | 1.35e-4 | 9.57e-6 | 2.28e-5 | 3.21e-5 | 9.46e-5 | 4.58e-5 |
| Delta | 5.45e-6 | 2.53e-6 | 6.67e-6 | 2.37e-6 | 9.74e-6 | 1.44e-5 | 8.32e-6 | 9.78e-6 |
| KNN | 9.07e-6 | 3.49e-6 | 9.37e-6 | 9.56e-6 | 4.88e-6 | 4.27e-6 | 3.39e-6 | 1.37e-5 |
| KNN+ | 9.11e-6 | 1.28e-5 | 1.01e-5 | 3.44e-6 | 1.21e-6 | 4.58e-6 | 1.05e-5 | 2.49e-5 |
| Ours | **1.21e-6** | **1.45e-6** | **2.03e-6** | **7.63e-7** | **6.56e-7** | **1.04e-6** | **8.88e-7** | **2.02e-6** |

Table 10: Comparison of MSEs for adding and removing one data point, using MC-estimated values as ground truth. The data values estimated by our proposed methods, IncGLOC and DecGLOC, achieve the lowest MSEs, indicating a closer approximation to the true Shapley values.

dataset and previously estimated values, remains effective and appropriate. However, in cases of substantial dataset changes, it is preferable to recompute data values using the proposed GLOC method. Alternatively, a step-wise strategy can be employed, in which no more than 10% of the dataset is modified at each step, and IncGLOC or DecGLOC is applied iteratively to perform incremental or decremental valuation.

## A.8 INTEGRATION WITH OTHER DATA VALUATION METHODS

The proposed regularization terms regarding value distributions can be seamlessly incorporated into various valuation frameworks. These regularizers can be integrated either concurrently with existing valuation methods or as a post-processing step. The first strategy, where our regularizers are applied alongside another valuation approach, has been exemplified using the AME method. Furthermore, we demonstrate that these regularization terms can also function as standalone optimization objectives to refine the data values derived from other valuation techniques. Table 5 presents the performance of the original data valuation methods alongside their refined values after incorporating our proposed objectives. The results indicate that incorporating the global and local distribution characteristics of value distributions can further improve the accuracy of the data values obtained through our valuation methods, thereby enhancing detection performance.

| Manner | Add | | | | Remove | | | |
|---|---|---|---|---|---|---|---|---|
| Dataset | Electricity | MiniBooNE | CIFAR10 | Fried | Electricity | MiniBooNE | CIFAR10 | Fried |
| MC | 33:1 | 40:1 | 14:1 | 12:1 | 8:1 | 3:1 | 19:1 | 5:1 |
| TMC | 506:1 | 63:1 | 149:1 | 6:1 | 47:1 | 30:1 | 226:1 | 151:1 |
| Delta | 4:1 | 2:1 | 3:1 | 2:1 | 41:1 | 18:1 | 11:1 | 6:1 |
| KNN | 22:1 | 3:1 | 7:1 | 25:1 | 8:1 | 3:1 | 3:1 | 19:1 |
| KNN+ | 20:1 | 23:1 | 16:1 | 3:1 | 3:1 | 3:1 | 15:1 | 20:1 |

Table 11: Ratios of MSEs between the compared baselines and our approach for adding and removing a single data point, using AME estimates as a proxy for the true Shapley values and MC estimates as initial values. Our method achieves improvements in Shapley value estimation accuracy ranging from at least $2\times$ to as much as $506\times$ compared to baseline approaches.

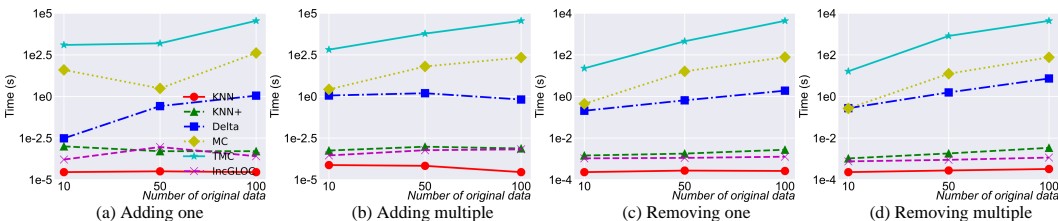

(a) Adding one  (b) Adding multiple  (c) Removing one  (d) Removing multiple

Figure 9: Comparison of the computational costs between IncGLOC, DecGLOC, and other baseline methods for adding or removing both single and multiple data points. Methods that do not require re-estimating the Shapley values, such as KNN, KNN+, and our proposed methods (i.e., IncGLOC and DecGLOC), consistently demonstrate superior computational efficiency.

| Metric | Electricity | MiniBooNE | CIFAR10 | BBC | Fried | 2Dplanes | Pol | Covertype | Nomao | Law | Creditcard | Jannis |
|--------|-------------|-----------|---------|-----|-------|----------|-----|-----------|-------|-----|------------|--------|
| Cosine | **50:1** | **8:1** | **96:1** | **6:1** | **82:1** | **105:1** | **7:1** | **113:1** | **44:1** | **18:1** | **54:1** | **206:1** |
| Euclidean | 28:1 | 5:1 | 84:1 | 5:1 | 35:1 | 70:1 | 6:1 | 103:1 | 38:1 | 15:1 | 52:1 | 127:1 |

Table 12: Performance comparison of different similarity metrics for Shapley value estimation.

## A.9 MORE EXPERIMENTS FOR VALUE-BASED POINT ADDITION AND REMOVAL

We present additional results on value-based data point addition and removal experiments. Fig. 7 illustrates the test accuracy curves for the point removal experiment. Among the evaluated methods, GLOC generally exhibits the most significant performance degradation, suggesting its effectiveness in identifying high-quality samples. Notably, for the Electricity and MiniBooNE datasets, DVRL also performs well; however, its effectiveness is considerably lower on the other two datasets. Fig. 8 shows the test accuracy curves for the point addition experiment. When only low-quality samples are added, GLOC demonstrates a substantial decline in performance, indicating its capability to detect and differentiate poisoned samples. These findings collectively validate the effectiveness and reliability of GLOC in data valuation.

Additionally, we conduct experiments comparing GLOC and AME in scenarios involving the removal of low-value data points and the addition of high-value ones. Due to its higher valuation accuracy, GLOC is more effective at distinguishing high-quality from low-quality samples. Tables 6 and 7 present the comparison between GLOC and AME for these two scenarios. Ideally, removing the least valuable samples or adding the most informative ones should enhance model performance under noisy conditions. The results demonstrate that GLOC consistently outperforms AME in both cases.

## A.10 MORE EXPERIMENTS FOR MISLABELED DATA DETECTION

We present additional comparative results for the mislabeled data detection task. Table 8 reports the F1-scores of various data valuation approaches across six classification datasets with 10% noise. Although GLOC is adapted from AME, which typically exhibits suboptimal performance in mislabeled data detection, our proposed GLOC approach demonstrates state-of-the-art performance in noise detection tasks, outperforming all compared baselines.

## A.11 MORE EXPERIMENTS FOR DYNAMIC DATA VALUATION

We conduct additional experiments to assess the performance of IncGLOC and DecGLOC in the context of dynamic data valuation. While the results for adding or removing a single data point

| Dataset | Pol | Jannis | Law | Covertype | Nomao | Creditcard |
|---------|-----|--------|-----|-----------|-------|------------|
| Cosine | **0.66±0.009** | **0.30±0.007** | **0.96±0.008** | **0.53±0.011** | **0.68±0.006** | **0.46±0.005** |
| Euclidean | 0.61±0.005 | 0.27±0.006 | 0.95±0.010 | 0.51±0.007 | 0.67±0.012 | 0.44±0.006 |

Table 13: Performance comparison of different similarity metrics for mislabeled data detection.

| Dataset | BBC | CIFAR10 | Law | Electricity | Fried |
|---------|-----|---------|-----|-------------|-------|
| AME | 21.45s | 489.54s | 7.24s | 8.32s | 10.48s |
| GLOC | 21.58s | 490.13s | 7.25s | 8.34s | 10.53s |

Table 14: Comparison of computational time between GLOC and AME.

| Dataset | Electricity | MiniBooNE | CIFAR10 | BBC |
|---------|-------------|-----------|---------|-----|
| GLOC | **0.86e-6** | **0.75e-6** | **1.43e-5** | **1.75e-6** |
| $-\mathcal{R}_g$ | 0.96e-5 | 1.13e-6 | 2.44e-4 | 5.82e-6 |
| $-\mathcal{R}_l$ | 1.42e-5 | 1.27e-6 | 2.92e-4 | 6.50e-6 |

Table 15: Ablation studies on the two regularization terms, namely $\mathcal{R}_g$ and $\mathcal{R}_l$, in Shapley value estimation across four datasets.

are presented in the main text, the outcomes for adding or removing two data points are provided in Table 9. The proposed methods, IncGLOC and DecGLOC, consistently yield the lowest MSEs, underscoring their effectiveness in data valuation within dynamic data scenarios.

Furthermore, we evaluate the performance of various methods using MC-estimated values, where the number of sampled subsets is set equal to the total number of training samples, to serve as the ground-truth Shapley values. As in the main text, except for MC and TMC, which compute data values from scratch, all other methods are initialized with MC-estimated values. The compared methods are summarized in Table 10. The results show that our method consistently achieves superior performance even under this revised evaluation protocol.

Additionally, to highlight that our proposed methods yield substantial relative gains over baseline approaches despite the small absolute MSE values, Table 1 in the main text reports the MSE ratio between AME and GLOC, with the maximum improvement reaching 206-fold. Moreover, Table 11 reports the MSE ratios (rounded to integers) between baseline methods and our approaches under the dynamic data valuation setting, where AME estimates are used as a proxy for the true Shapley values and MC estimates serve as initial values. The results indicate that our methods, including IncGLOC and DecGLOC, attain improvements ranging from at least $2\times$ up to $506\times$ in Shapley value estimation accuracy compared to other baseline approaches.

## A.12 DIFFERENT SIMILARITY METRICS

In our approach, cosine similarity is employed as the similarity metric. This section compares model performance using both cosine similarity and Euclidean distance. Under Euclidean distance, $\mathcal{S}_{i,j}$ is defined as $\mathcal{S}_{i,j} = \frac{1}{d_E(\boldsymbol{x}_i, \boldsymbol{x}_j)} \cdot [2\mathcal{I}(y_i = y_j) - 1]$. The comparisons are conducted under two settings: Shapley value estimation and mislabeled data detection. Here, $d_E(\boldsymbol{x}_i, \boldsymbol{x}_j)$ denotes the distance between the features of samples $\boldsymbol{x}_i$ and $\boldsymbol{x}_j$. As shown in Tables 12 and 13, cosine similarity consistently outperforms Euclidean distance, likely due to its ability to capture directional alignment while ignoring magnitude, thereby better reflecting the semantic similarity between data points.

## A.13 COMPUTATIONAL EFFICIENCY

First, we compare the computational efficiency of our approach with AME on five datasets. Table 14 presents the computation time of the proposed GLOC method, which is built upon AME, relative to the original AME. The results show that GLOC, by incorporating only two distribution-aware

| Dataset | CIFAR10 | Fried | 2Dplanes | Covertype | Creditcard | Jannis |
|---------|---------|-------|----------|-----------|------------|--------|
| GLOC | **96:1** | **82:1** | **105:1** | **113:1** | **54:1** | **206:1** |
| $-\mathcal{R}_g$ | 6:1 | 21:1 | 34:1 | 14:1 | 20:1 | 68:1 |
| $-\mathcal{R}_l$ | 5:1 | 17:1 | 12:1 | 9:1 | 22:1 | 35:1 |

Table 16: Ablation study of MSE ratios between AME and GLOC for Shapley value estimation, assessing the impact of the two regularization terms, $\mathcal{R}_g$ and $\mathcal{R}_l$.

| Metric | Dataset | AME | GLOC | $-\mathcal{R}_g$ | $-\mathcal{R}_l$ |
|--------|---------|-----|------|------------------|------------------|
| MAE | Electricity | 5.23e-3 | **7.30e-4** | 2.47e-3 | 3.00e-3 |
| | BBC | 2.61e-3 | **1.06e-3** | 1.92e-3 | 2.01e-3 |
| Spearman | Electricity | 0.68 | **0.85** | 0.80 | 0.75 |
| | BBC | 0.72 | **0.81** | 0.75 | 0.77 |

Table 17: Ablation study of the two regularization terms using MAE and the Spearman correlation between the estimated data values and the ground truth for Shapley value estimation.

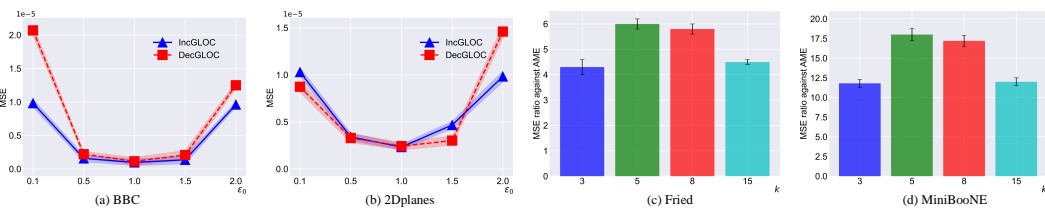

Figure 10: (a) and (b) Sensitivity tests for the value of $\epsilon_0$ under both incremental and decremental data valuation scenarios. The average performance for adding or removing one and two data points is reported. (c) and (d) Sensitivity tests for the value of the neighborhood size $k$.

regularization terms into AME, introduces minimal additional training time while enhancing Shapley value estimation accuracy by up to 206-fold (Table 1).

We compare the computational efficiency of our dynamic data valuation methods, including IncGLOC and DecGLOC, with baseline approaches, as shown in Fig. 9. Methods that do not require re-estimating the Shapley values, such as KNN, KNN+, and our proposed approaches, demonstrate superior efficiency. In contrast, methods that necessitate recalculating Shapley values, such as MC and TMC, incur significantly higher computational costs for dynamic data valuation, even when adding or removing a single data point.

## A.14 FURTHER ABLATION STUDIES ON THE TWO REGULARIZATION TERMS

Tables 15 and 16 present additional results from the ablation studies on the regularization terms in GLOC. When both the global and local regularization terms are incorporated, the estimation performance of the data values is maximized, emphasizing the critical role of integrating global and local statistical information for accurate and effective data valuation. Moreover, we employ additional metrics to conduct ablation studies on the proposed two regularization terms, including MAE and Spearman correlation between the estimated data values and their ground truth. The results for these metrics are presented in Table 17. From the results, when one regularization term is removed, GLOC's performance declines but still consistently outperforms AME, highlighting the necessity of each regularization term. Moreover, despite the significant difference in sample sizes between Electricity (38,474 samples) and BBC (2,225 samples), the performance trends remain consistent. Moreover, Spearman correlation complements MSE/MAE-based evaluation by capturing consistency in relative value rankings. Notably, GLOC estimates show the highest correlation with the ground-truth Shapley values, further validating its effectiveness in achieving accurate data valuation.

| Ratio | 0.2 | 0.4 | 0.6 | 0.8 |
|-------|-----|-----|-----|-----|
| GLOC | **0.786** | **0.764** | **0.763** | **0.742** |
| $-\mathcal{R}_g$ | 0.790 | 0.781 | 0.769 | 0.743 |
| $-\mathcal{R}_l$ | 0.792 | 0.776 | 0.774 | 0.751 |
| AME | 0.809 | 0.799 | 0.785 | 0.768 |

Table 18: Ablation study on the Fried dataset assessing accuracy under varying proportions of data removal, examining the impact of the two regularization terms, $\mathcal{R}_g$ and $\mathcal{R}_l$.

| Ratio | 0.05 | 0.1 | 0.15 | 0.2 |
|---|---|---|---|---|
| GLOC | **0.658** | **0.613** | **0.527** | **0.394** |
| $-\mathcal{R}_g$ | 0.662 | 0.615 | 0.549 | 0.428 |
| $-\mathcal{R}_l$ | 0.680 | 0.622 | 0.560 | 0.451 |
| AME | 0.714 | 0.647 | 0.640 | 0.637 |

Table 19: Ablation study on the MiniBooNE dataset evaluating accuracy under varying proportions of data addition, investigating the effects of the two regularization terms, $\mathcal{R}_g$ and $\mathcal{R}_l$.

| $k$ | 3 | 5 | 8 | 10 | 15 |
|---|---|---|---|---|---|
| Electricity | 27:1 | **50:1** | 46:1 | 45:1 | 30:1 |
| CIFAR10 | 53:1 | **96:1** | 95:1 | 87:1 | 48:1 |
| 2Dplanes | 58:1 | **105:1** | **105:1** | 101:1 | 64:1 |
| BBC | 4:1 | 6:1 | **8:1** | 5:1 | 4:1 |

Table 20: Sensitivity analysis of the hyperparameter $k$ in the GLOC approach.

Additionally, we conduct ablation studies under the value-based point addition and removal setting, where the global and local regularization terms are independently removed. From the results presented in Tables 18 and 19, removing either term leads to a slower rate of performance degradation. If rapid accuracy changes are primarily driven by neighborhood consistency regularization, which causes similar samples to be removed together, removing the global term should accelerate the performance drop. Thus, we conclude that the primary driver of the rapid accuracy change is the enhanced Shapley value estimation, facilitated by the synergistic effect of both global and local regularization terms.

## A.15 SENSITIVITY ANALYSES FOR HYPERPARAMETERS

We conduct sensitivity analyses on the hyperparameter $\epsilon_0$. The results presented in Figs. 10(a) and (b) show that the performance of our proposed dynamic data valuation methods, including IncGLOC and DecGLOC, remains stable when $\epsilon_0 \in [0.5, 1.5]$. Furthermore, we conduct sensitivity analyses on the neighborhood size used in the proposed regularization term to capture local distribution characteristics. The results presented in Figs. 10(c) and (d) demonstrate that the model achieves optimal performance when $k$ is set to 5 for the incremental data valuation. Additionally, sensitivity tests on the value of $k$ are also conducted for the GLOC approach. As shown in Table 20, consistent with IncGLOC, optimal performance is achieved at $k = 5$ or $k = 8$ across various datasets, indicating these as recommended default settings.

## A.16 DETERMINATION OF THE PERMISSIBLE VARIATION BOUND

We conduct ablation studies to investigate the permitted variation bound of data values under dynamic data valuation scenarios. Three settings are considered. In Setting I, the bound is determined solely by the variation within the dataset. In Setting II, the bound is based exclusively on the variation within the sample's neighborhood. In Setting III, the bound is determined by both the variation within the entire dataset and the variation within the neighborhoods of the samples. The results presented in Table 21 demonstrate that the optimal performance is achieved when both variations are taken into account. Furthermore, the findings suggest that local distribution characteristics play a more critical role than global information in determining the variation bound of data values.

## A.17 DISCUSSIONS & LIMITATIONS

While the proposed global and local distribution-aided data valuation methods exhibit strong performance, they also have certain limitations that present avenues for future research. First, our study primarily focuses on global and local distributions of data values. Future research could extend our analysis by exploring value distributions from alternative perspectives, such as hierarchical and conditional distributions of data values, to further enhance the generalizability and robustness of our approach. Moreover, although the proposed dynamic data valuation method avoids repeated Shapley value computations, its performance remains dependent on the quality of existing data values. To

| Manner | Add | | | | Remove | | | |
|---|---|---|---|---|---|---|---|---|
| Dataset | Electricity | MiniBooNE | CIFAR10 | Fried | Electricity | MiniBooNE | CIFAR10 | Fried |
| Setting I | 6.73e-6 | 4.99e-6 | 6.29e-6 | 3.54e-6 | 2.53e-6 | 4.27e-6 | 4.05e-6 | 8.55e-6 |
| Setting II | 2.13e-6 | 2.87e-6 | 5.01e-6 | 2.79e-6 | 1.66e-6 | 2.83e-6 | 4.15e-6 | 3.37e-6 |
| Setting III | **1.73e-6** | **1.99e-6** | **3.29e-6** | **2.17e-6** | **0.95e-6** | **2.00e-6** | **2.55e-6** | **2.27e-6** |

Table 21: Ablation study on the permissible variation bound of data values in dynamic scenarios.

further enhance effectiveness, one potential improvement is to first refine the existing values using the proposed regularization terms prior to applying dynamic data valuation. This additional step could improve both accuracy and robustness in the valuation process.

Furthermore, the reliance of our approach on neighboring data points may limit privacy guarantees. Future work could investigate privacy-preserving extensions, for instance, via differential privacy or local perturbation techniques. Additionally, despite the strong practical utility of our approach, similar to AME and other existing scalable valuation frameworks, it does not strictly enforce the Shapley efficiency axiom. Developing methods that better satisfy this axiom while maintaining efficiency presents another promising avenue for future work. Finally, our experiments cover tasks including Shapley value estimation, value-based data addition and removal, mislabeled data detection, and dynamic data valuation. Future research should explore the applicability of our approach to more complex data scenarios, such as cross-modal data and non-i.i.d. data.

## A.18 ETHICAL STATEMENT

All models and datasets used in this study have been meticulously processed and curated by their respective publishers to mitigate any ethical issues.

## A.19 REPRODUCIBILITY STATEMENT

Our code is available in the supplementary material, and all the datasets used are publicly available.

## A.20 USAGE OF LARGE LANGUAGE MODELS

The authors declare that large language models were not employed in the ideation or writing of the research. They were used exclusively to identify and correct grammatical errors for partial sentences.

