# OpenReview forum: "Data Valuation by Leveraging Global and Local Statistical Information"
_ICLR.cc/2026/Conference — ICLR 2026 Conference Withdrawn Submission_

### Official Review · Reviewer_zxMX · 2025-10-28

**Soundness:** 3
**Presentation:** 4
**Contribution:** 3
**Rating:** 6
**Confidence:** 3

**Summary:**

This paper proposes Global and LOcal Characteristics-based (GLOC) data valuation, an improved approach for efficient data valuation built upon an earlier method called the Average Marginal Effect (AME)  (Lin et al., 2022) or Shapley value approach. Instead of relying on Lasso regression, the authors modify the objective with two key enhancements: a global Gaussian-based term and a local neighborhood-consistency term. The global term replaces the original Laplace prior with an L2 norm, motivated by the empirical observation that data Shapley values more closely follow a Gaussian rather than Laplace distribution. The local term enforces that data points with similar features and labels should have more consistent values, refining Shapley estimation through neighborhood regularization. Additionally, the paper develops approaches to allow efficient updates when data points are added or removed without recomputing Shapley values from scratch. They conduct experiments across multiple tasks and justify the performance GLOC.

**Strengths:**

1. The proposed approach is simple and efficient, yet appears effective in experiments. I liked the insights about the empirical distribution observations about Shapley values and how that help us to refine AME approach.
2. The empirical evaluation is thorough, with different datasets, different tasks and comparisons with multiple natural baselines. The implementation details are also nicely described.
3. Overall writing of the paper is great.

**Weaknesses:**

I generally liked the paper and its effort. The only limitation is that while the methods seem empirically effective, the underlying idea (both the change of prior distribution and regularization) appears a bit incremental, though I do not see this as a significant limitation.

**Questions:**

N/A

---

### Official Review · Reviewer_NiFP · 2025-10-31

**Soundness:** 3
**Presentation:** 3
**Contribution:** 3
**Rating:** 6
**Confidence:** 5

**Summary:**

This paper proposes a novel, distribution-aware perspective for data valuation. The authors identify key global and local statistical characteristics of the data values and introduce a new valuation method GLOC. Furthermore, the paper proposes an efficient dynamic valuation framework (IncGLOC/DecGLOC) capable of updating data values without full Shapley value recalculation. Comprehensive experiments across multiple tasks validate the method's effectiveness and efficiency.

**Strengths:**

1. The paper presents a novel premise. It explores the "value distribution" dimension, a concept largely overlooked by prior work, offering a new approach to regularizing the ill-posed problem of SV estimation.
2. The local regularization term in GLOC effectively encodes value consistency. This design is not only sound but also easily integrated into other data valuation frameworks, demonstrating good extensibility.
3. The evaluation is comprehensive, covering both proxy SV estimation and multiple downstream tasks (e.g., mislabeled data detection, data addition/removal) that measure practical utility. The comparison against a wide array of baselines is commendable.

**Weaknesses:**

1. The dynamic valuation method (e.g., the $\epsilon_i$ bound) is effective but based on heuristics. It currently lacks a theoretical justification or convergence analysis.
2. The authors claim their method is an approximate method of Shapley value, but the added regularization term violates axioms such as efficiency, symmetry, and additivity. The paper does not prove whether or how these properties are preserved.
3. The Shapley estimation experiment compares only against AME, which is itself a weak Monte Carlo baseline.

**Questions:**

1. Section 4.2 should be updated to explain how the GLOC objective (Eq. 5) remains stable in the under-determined $M \ll N$ setting. The relative importance of $R_g$ vs. $R_l$ as stabilizers should be discussed.
2. The metrics focus on task accuracy after removing/adding points, which reflect ranking utility, not estimation fidelity. The paper should report MSE/MAE/Spearman/ correlation against the exact Shapley value on enumerable datasets.
3. The definition of the neighborhood similarity metric S_{i,j} and the variation bound \epsilon_i involves multiple heuristic constants without sensitivity analysis.

---

### Official Review · Reviewer_HLw3 · 2025-10-31

**Soundness:** 1
**Presentation:** 3
**Contribution:** 2
**Rating:** 2
**Confidence:** 4

**Summary:**

Based on the empirical observation of local and global statistics of data values computed by AME, the paper proposes two regularizers, which appear in Eq. (5), that use the corresponding statistics as priors to expedite the computation of data values. Intuitively, these two regularizers may be used for other optimization-based data valuation methods. Moreover, the authors also empirically study the statistics before and after adding or deleting data, and then develop an optimization, as defined in Eq. (8), to efficiently adjust the computed data values accordingly.

**Strengths:**

- The authors conduct extensive experiments to support their claims.

- I think it is a sensible approach to start from empirical observations.

**Weaknesses:**

- I am concerned with the use of AME for approximating the Shapley value. As far as I know, there is no theoretical guarantee that the solution of AME would converge to the Shapley value. In theory, I do not see this is possible. The underlying $p$ for the Shapley value is the uniform distribution over the closed interval $[0,1]$, and then the value of $v$ defined in 257 is equal to infinity. Notice that $v$ is used in constructing $\mathbf{X}$ in Eqs. (3) and (5), which makes AME infeasible. To circumvent this issue, the AME paper proposes to use the uniform distribution over $[\epsilon, 1-\epsilon]$ instead. However, this would unnecessarily introduce some bias. The provided results would be more convincing if the authors use KernelSHAP instead, which has theoretical guarantees on its convergence to the Shapley value; see [1].

   - In particular, the used distribution of $p$ is provided in line 901, which is $\mathrm{Uniform}\\{0.2, 0.4, 0.6, 0.8\\}$. This indicates that the solution of AME converges to $\boldsymbol\omega^*$ defined as

        $$
        \omega\_{i}\^{*} = \sum\_{S \subseteq [N]\setminus \\{i\\}} \frac{1}{4}\left( \sum\_{w\in\\{0.2,0.4,0.6,0.8\\}} w\^{|S|}(1-w)\^{N-|S|-1}\right)\cdot[U(S\cup\\{i\\}) - U(S)]
        $$

        where $U$ denotes the utility function used to construct $\mathcal{U}$ in Eq. (3). Clearly, this is not the Shapley value. **Therefore, it is remarkably erroneous that the paper claims the Shapley value as the ground truth throughout.**

- **Theoretically, the imposed regularizers shift the convergence point, which makes the empirical MSE results (e.g., Table 1) questionable**. Their claims make it seem as if the performance boost in value-based point addition and removal comes from better approximating some ground truth, but this is misleading.

- In lines 340-341, I do not think that $|\hat{\mathcal{D}}|/|\mathcal{D}|$ is a reasonable quantity that measures the variation brought by adding $\mathcal{D}'$. For example, if $\mathcal{D}'$ is constructed by perturbing all data in $\mathcal{D}$ a little bit, making $\mathcal{D}'$ almost a copy of $\mathcal{D}$ but not the same, then I do not see it would introduce variation.


Overall, the paper notably lacks coherence, as its theoretical justification does not align with the empirical results or the authors’ claims.

**Not weaknesses**

- It looks like that the definition of $\overline{\epsilon}$ in Eq. (8) is missing in the main paper?

- In line 364-365, the authors may intend to mean ''not only for AME'' instead of "except for AME?"

**References**

[1] Chen, T., Seshadri, A., Villani, M. J., Niroula, P., Chakrabarti, S., Ray, A., ... & Kumar, N. (2025). A Unified Framework for Provably Efficient Algorithms to Estimate Shapley Values. In NeurIPS 2025.

**Questions:**

-  Why the two introduced regularizers would help expedite the convergence in theory?

---

### Official Review · Reviewer_fPii · 2025-11-01

**Soundness:** 3
**Presentation:** 3
**Contribution:** 2
**Rating:** 4
**Confidence:** 3

**Summary:**

This paper proposes an enhanced Shapley-value-based data valuation approach that leverages both global and local statistical properties of data value distributions. The authors introduce new regularization terms into an existing valuation framework  to incorporate distribution characteristics, aiming to refine the estimation of each data point’s value. They further present a dynamic data valuation technique that efficiently updates data values when the dataset changes  without recomputing Shapley values from scratch. Through experiments on various tasks, the paper claims improved performance and efficiency over baseline methods, demonstrating the potential benefits of using global and local value distribution information in data valuation.

**Strengths:**

- The idea of integrating global and local value distribution statistics into data valuation is novel.
- The proposed method is easy to implement. It integrates with existing AME pipelines, and scales via convex solvers
- The paper provides extensive experimental validation across multiple datasets and tasks.

**Weaknesses:**

- The core conceptual novelty is a new regularizer and an inference procedure, which is modest and would be strengthened by stronger theoretical analysis. Although the idea is well-motivated by empirical observations  especially when recalculating Shapley values is expensive, but the combination of regularization and constrained quadratic inference is somewaht incremental
- Inc/DecGLOC are heuristic but reasonable. The new regularizers and dynamic update rules appear to be designed empirically based on observed patterns, which raises concerns about how generally they will apply beyond the tested cases. Clarifying the rationale behind these specific design choices or deriving them from a theory would strengthen the contribution.
- Much of the analysis uses AME estimates and sometimes AME is used for initial values, which risks propagating AME biases into conclusions. It would be helpful to validate the Gaussian and neighborhood claims against exact Shapley values on small datasets  or against Monte Carlo estimates with very large sampling.

**Questions:**

- What is the justification for the specific regularizers and distribution assumptions chosen?  It would help to clarify why these forms  are expected to improve data value estimates in general.
- can the authors provide arguments or proofs about convergence to true Shapley values, or conditions under which the regularized estimation remains accurate?
- Can the authors show exact-Shapley-based distributions for a few small datasets  to confirm that true Shapley values  are approximately Gaussian?

---

### Official Review · Reviewer_TjRT · 2025-11-02

**Soundness:** 2
**Presentation:** 3
**Contribution:** 2
**Rating:** 2
**Confidence:** 3

**Summary:**

This paper proposes **GLOC**—an AME-style regression objective enhanced with two *distribution-aware* regularizers: a **global Gaussian prior** (L2/Ridge instead of AME’s L1/Laplace) and a **local neighborhood-consistency** term (same-class neighbors closer; different-class neighbors separated). It also presents **IncGLOC/DecGLOC** for **dynamic** scenarios, updating values after data addition/removal **without recomputing Shapley**. Across 12 datasets, under a unified **logistic-regression** surrogate (pretrained embeddings for text/image), GLOC reduces MSE to Shapley references and improves value-guided curation and mislabeled/noisy detection; Inc/DecGLOC are competitive and more efficient for dynamics.

**Strengths:**

- **Motivated & simple:** Empirical findings (“global ≈ Gaussian; local same-class consistency”) are encoded as two lightweight regularizers atop AME (Ridge + neighborhood Laplacian).
- **Dynamic efficiency:** Inc/DecGLOC infer updated values from the new dataset and prior values without recomputing Shapley, reducing time in add/remove settings.
- **Clean protocol:** A fixed logistic-regression surrogate (pretrained DistilBERT/ResNet50 for features) makes cross-method comparison reproducible.
- **Consistent gains:** Lower MSE to Shapley references; better value-based add/remove and mislabeled detection on multiple datasets; useful ablations (Rg/Rl, k, ε₀) and similarity choices.
- **Modular:** The regularizers are shown to be usable beyond AME (integration or post-refinement).

**Weaknesses:**

- **Novelty is modest (accurate statement):** The main contributions are *regularizing* AME with a Gaussian prior (L2) and a neighborhood-consistency term, plus a heuristic dynamic update; conceptually this is an **incremental** extension rather than a new valuation principle.
- **No end-to-end deep validation:** Experiments use a **logistic-regression** surrogate with **pretrained** DistilBERT/ResNet50 **feature extraction**; there is no end-to-end CNN/Transformer training, which may limit claims about generality to deep models.
- **Limited formal guarantees for dynamics:** The dynamic method sets the per-point change bound \( \epsilon_i \) **heuristically** (function of dataset/neighbor changes) and does **not** provide formal error/stability bounds vs. recomputing Shapley.
- **Ground-truth choice for MSE:** For static estimation, “ground-truth Shapley” is proxied by **AME with very large M** (equal to training size). While standard in this line, it is still a proxy rather than exact Shapley.

**Questions:**

Same with weakness

---

### Note · Authors · 2025-11-27

I have read and agree with the venue's withdrawal policy on behalf of myself and my co-authors.